# Asynchronous Multi-Agent Reinforcement Learning with General Function Approximation

## Abstract

We study multi-agent reinforcement learning (RL) where agents cooperate through asynchronous communications with a central server to learn a shared environment. Our first focus is on the case of multi-agent contextual bandits with general function approximation, for which we introduce the Async-NLin-UCB algorithm. This algorithm is proven to achieve a regret of $\widetilde{O}(\sqrt{T \dim_E(\mathcal{F}) \log N(\mathcal{F})})$ and a communication complexity of $\widetilde{O}(M^2 \dim_E(\mathcal{F}))$, where $M$ is the total number of agents and $T$ is the number of rounds, while $\dim_E(\mathcal{F})$ and $N(\mathcal{F})$ are the Eluder dimension and the covering number of function space $\mathcal{F}$ respectively. We then progress to the more intricate setting of multi-agent RL with general function approximation, and present the Async-NLSVI-UCB algorithm. This algorithm enjoys a regret of $\widetilde{O}(H^2 \sqrt{K \dim_E(\mathcal{F}) \log N(\mathcal{F})})$ and a communication complexity of $\widetilde{O}(HM^2 \dim_E(\mathcal{F}))$, where $H$ is the horizon length and $K$ the number of episodes. Our findings showcase the provable efficiency of both algorithms for collaborative learning within nonlinear environments and minimal communication overhead.

## 1 Introduction

Multi-agent reinforcement learning (RL) is an important paradigm in RL, and has been successfully applied to real-world tasks such as robotics [Williams et al., 2016, Liu et al., 2019, Ding et al., 2020, Liu et al., 2020, Na et al., 2022], games [Vinyals et al., 2017, Berner et al., 2019, Jaderberg et al., 2019, Ye et al., 2020], and control systems [Bazzan, 2009, Yu et al., 2014, 2020, Min et al., 2022, Xu et al., 2023]. By learning cooperatively, agents benefit from sharing learning experiences, enabling them to collectively enhance their decision-making capabilities. This collaborative process is usually accomplished through the utilization of a central server, whose task is to aggregate local data and deliver feedback for the agents.

There has been an excellent line of work establishing provably efficient algorithms for multi-agent bandits and RL. However, most existing works are restricted to the synchronous setting, where communications between all agents and the server must happen simultaneously. This is impractical since in many scenarios the availability of agents may vary and be unpredictable. Ideally, communication should be allowed to happen asynchronously to offer the agents more flexibility. He et al. [2022] and Min et al. [2023] studied this setting respectively for *linear contextual bandits* and *linear Markov Decision Processes* (MDPs), both of which assumes linearity in the environment, and introduced algorithms with low regret and communication cost. Yet the linear function class is quite limited, and does not encompass practical reinforcement learning scenarios where nonlinearity is prevalent.

To address the aforementioned drawback, in this work, we tackle environments with general function approximation, broadening the applicability of the algorithm to more realistic and complex scenarios. We first delve into multi-agent contextual bandits with general function approximation, where multiple agents interact with homogeneous environments in parallel to solve a common objective. Notably, the communication protocol is designed to be flexible and asynchronous, allowing agents to initiate communication with the server and acquire new policy functions whenever the need arises. The primary objective is to minimize total regret while reducing communication cost as much as possible.

Submitted to 38th Conference on Neural Information Processing Systems (NeurIPS 2024). Do not distribute.

We propose an algorithm Async-NLin-UCB, which adapts a fully asynchronous communication protocol, and leverages various methods for tackling nonlinear function approximation. Despite the flexibility of communication, our algorithm performs almost as well as a single agent, in terms of a regret that is mostly independent of the number of agents and a low communication cost.

We then progress to multi-agent RL with general function approximation under similar requirements and objectives. We propose an algorithm named Async-NLSVI-UCB based on Least-Squares Value Iteration (LSVI) to learn the underlying Markov decision processes (MDPs), which demonstrates similar advantages with provably low regret and communication cost.

Our main contributions are summarized in the following:

- For asynchronous multi-agent nonlinear contextual bandits, we propose the algorithm Async-NLin-UCB, which enjoys an $\widetilde{O}(\sqrt{T \dim_E(\mathcal{F}) \log N(\mathcal{F})} + \dim_E(\mathcal{F}))$ regret and an $\widetilde{O}(M^2 \dim_E(\mathcal{F}))$ communication complexity, where $\dim_E(\mathcal{F})$ and $N(\mathcal{F})$ are respectively the Eluder dimension and the covering number of function space $\mathcal{F}$.

- For asynchronous multi-agent nonlinear MDPs, we propose the algorithm Async-NLSVI-UCB, which enjoys an $\widetilde{O}(H^2 \sqrt{K \dim_E(\mathcal{F}) \log N(\mathcal{F})} + H^2 \dim_E(\mathcal{F}))$ regret and a communication complexity of $\widetilde{O}(HM^2 \dim_E(\mathcal{F}))$.

- At the core of our algorithm, we design a *communication criterion* in order to tackles the challenges posed by both asynchronous communication and the nonlinearity of function approximation. To guarantee a low communication cost, we propose a low switching communication criterion that allows the agent to trigger communication rounds.

- We carefully design our *download content* from server to local agents, which consist only of decision and bonus functions, with no mention of any specific historical data. This effectively protects user data against exposure by disallowing local users from obtaining the data of others.

**Notation.** We use lower case letters to denote scalars. We denote by $[n]$ the set $\{1, \ldots, n\}$. For two positive sequences $\{a_n\}$ and $\{b_n\}$ with $n = 1, 2, \ldots$, we write $a_n = O(b_n)$ if there exists an absolute constant $C > 0$ such that $a_n \leq Cb_n$ holds for all $n \geq 1$. We use $\widetilde{O}(\cdot)$ to further hide the polylogarithmic factors. For two non-negative integers $a, b$ satisfying $a < b$ and a sequence $\{s_i\}$ indexed by integers $i$, we use $s_{[a:b]}$ to denote the subsequence $\{s_a, s_{a+1}, \cdots, s_b\}$.

## 2 Related Work

### 2.1 Multi-Agent Bandits

First, there is a multitude of previous work on distributed or federated multi-armed bandits and stochastic linear bandits [Liu and Zhao, 2010, Szorenyi et al., 2013, Landgren et al., 2016, Chakraborty et al., 2017, Landgren et al., 2018, Martínez-Rubio et al., 2019, Sankararaman et al., 2019, Wang et al., 2020a,c, Zhu et al., 2021, Huang et al., 2021]. For the more realistic setting of contextual bandits, most previous work are within the scope of linear contextual bandits with synchronized communication. Korda et al. [2016] introduced two novel distributed confidence ball (DCB) algorithms for linear bandit problems in peer-to-peer networks. Wang et al. [2020c] considered both P2P and star-shaped communication, achieving near-optimal regret and low communication cost that is largely independent of the time horizon in their algorithm DisLinUCB. Dubey and Pentland [2020] proposed FedUCB, an algorithm focusing on differential-privacy.

Li and Wang [2022] first considered an asynchronous communication protocol and proposed the algorithm Async-LinUCB with near-optimal regret, yet the algorithm contains a download step for all agents triggered by the central server. Their results are flexible and contains a parameter to control the trade-off between regret and communication cost. He et al. [2022] improved the setting to a fully asynchronous communication, proposing the algorithm FedLinUCB with near-optimal regret of $\widetilde{O}(d\sqrt{T})$ and low communication cost of $\widetilde{O}(dm^2)$, comparable to the benchmark in single-agent contextual linear bandits [Abbasi-Yadkori et al., 2011]. We consider the same communication protocol in our results. A summary of these results along with ours can be found in the first four rows of Table 1.

### 2.2 Multi-Agent RL

Multi-agent reinforcement learning is decidedly more challenging than contextual bandits. There is also a vast literature on this setting, with many works discussing different aspects of multi-agent RL

| Algorithm | Regret | Communication | Fully asynchrnous |
|-----------|--------|---------------|-------------------|
| `DisLinUCB` [Wang et al., 2020c] | $d\sqrt{MT}\log^2 T$ | $d^3 M^{3/2}$ | ✘ |
| `Async-LinUCB` [Li and Wang, 2022] | $dM^{(1-\gamma)/2}\sqrt{T}\log T$ | $dM^{1+\gamma}\log T$ | ✘ |
| `FedLinUCB` [He et al., 2022] | $d\sqrt{T}\log T$ | $dM^2\log T$ | ✓ |
| `Async-NLin-UCB` (ours) | $\sqrt{\dim_E \log N T}\log T$ | $\dim_E M^2 \log^2 T$ | ✓ |
| `Coop-LSVI` [Dubey and Pentland, 2021] | $d^{3/2}H^2\sqrt{MK}\log K$ | $dHM^3$ | ✘ |
| `Async-Coop-LSVI-UCB` [Min et al., 2023] | $d^{3/2}H^2\sqrt{K\log K}$ | $dHM^2\log K$ | ✓ |
| `Async-NLSVI-UCB` (ours) | $\sqrt{\dim_E \log N}H^2\sqrt{K}\log K$ | $\dim_E HM^2\log^2 K$ | ✓ |

Table 1: Comparison of our result against baseline methods for multi-agent contextual bandits and MDPs. Note that the first four rows are for contextual bandits, and the last three are for reinforcement learning. Only our algorithms are in the general function approximation setting. We abbreviate $\dim_E = \dim_E(\mathcal{F})$ and $N = N(\mathcal{F})$, and hide logarithmic factors. For algorithms with synchronized communication, each communication round actually corresponds to $M$ rounds in asynchronous settings, which explains the extra $M$ terms.

than ours. For example, there are works focusing on convergence guarantees [Zhang et al., 2018b,a, Wai et al., 2018], non-stationary or heterogeneous environments [Lowe et al., 2017, Yu et al., 2021, Dubey and Pentland, 2021, Kuba et al., 2022, Liu et al., 2022, Jin et al., 2022], and deep federated RL [Clemente et al., 2017, Espeholt et al., 2018, Horgan et al., 2018, Nair et al., 2015, Zhuo et al., 2019], to name a few. We refer to a recent survey on federated reinforcement learning Qi et al. [2021] for a more comprehensive summary.

Narrowing it down to multi-agent RL with function approximation, the benchmark is the LSVI-UCB algorithm in the single-agent setting [Jin et al., 2020], with an $\widetilde{O}(d^{3/2}H^2\sqrt{K})$ regret. Dubey and Pentland [2021] proposed CoopLSVI for multi-agent linear MDPs, which requires a synchronized communication through central server, and proves a regret of $\widetilde{O}(d^{3/2}H^2\sqrt{MK})$. They also extended their result to the heterogeneous setting. Min et al. [2023] considered the fully asynchronous setting and introduced the Async-Coop-LSVI-UCB algorithm, with a $\widetilde{O}(d^{3/2}H^2\sqrt{K})$ regret not dependent on the number of agents $M$, as well as a low communication cost. A summary of these results along with ours can be found in the last three rows of Table 1.

### 2.3 General function approximation

Reinforcement learning with general function approximation extends the well-studied case of linear MDPs to more general classes of MDPs, and has gained a lot of traction in recent years [Wang et al., 2020b, Jin et al., 2021, Foster et al., 2023, Du et al., 2021, Agarwal and Zhang, 2022, Agarwal et al., 2023]. Previous works focus on different measures of complexity for the function classes, for example the Bellman rank proposed by Jiang et al. [2017], the Bellman Eluder dimension introduced in Jin et al. [2021], the Decision-Estimation Coefficient in Foster et al. [2023], and generalized Eluder dimension in Agarwal et al. [2023]. Our work considers the Eluder dimension with the introduction of uncertainty estimators $D^2$, which has been widely utilized to establish results in RL with general function approximation [Agarwal et al., 2023, Zhao et al., 2023, Ye et al., 2023, Di et al., 2023].

## 3 Preliminaries

In this section, we introduce the formal definition of both multi-agent nonlinear contextual bandits and MDPs and some related concepts, and discuss the asynchronous communication protocol.

### 3.1 Multi-Agent Contextual Bandits with General Function Approximation

We assume a global action set $\mathcal{A}$ that is known to all agents. At each round $t \in [T]$, a single arbitrary agent $m_t \in [M]$ is chosen to participate. The agent receives a contextual decision set $\mathcal{A}_t \subseteq \mathcal{A}$ and chooses from the set an action $a_t \in \mathcal{A}_t$ to perform, and subsequently receives a random reward $r_t$.

The assumption of general function approximation is that the reward is generated according to

$$r_t = f^*(a_t) + \eta_t, \tag{1}$$

where $f^*$ is the ground truth objective function, and $\eta_t$ is a random noise variable. We assume the the objective function lies within a known function class $\mathcal{F}$. In addition, we also make the following assumptions regarding the function class and noise variables, which are standard assumptions for contextual bandits [Abbasi-Yadkori et al., 2011, He et al., 2022]:

**Assumption 3.1.** Suppose the following conditions hold for the contextual bandits environment:

- For any $f \in \mathcal{F}$ and $a \in \mathcal{A}$, $|f(a)| \leq 1$;

- $\eta_t$ is $R$-sub-Gaussian conditioned on data history: $\mathbb{E}\big[e^{\lambda \eta_t} \big| a_{1:t}, m_{1:t}, r_{1:t-1}\big] \leq \exp(R^2 \lambda^2 / 2), \forall \lambda$.

**Learning Objective.** The primary goal of contextual bandits is to minimize the cumulative regret

$$\text{Reg}(T) = \sum_{t=1}^{T} [f^*(a_t) - \max_{a \in \mathcal{A}_t} f^*(a)].$$

Notice that this summation is across all time steps does not depend on agent participation order, as should be the case for the resulting regret bound. To achieve this goal, agents are allowed to communicate with the server to upload their interaction history and update their policy. The secondary learning objective is to reduce communication overhead. We will explain the communication protocol further in Section 3.4.

### 3.2 Multi-Agent Episodic MDPs with General Function Approximation

We consider episodic MDPs, which are a classic family of models in reinforcement learning [Sutton and Barto, 2018]. It is characterized by the following elements, which we assume to be homogeneous across all agents: a state space $\mathcal{S}$, an action space $\mathcal{A}$, the horizon length $H$, transition probability functions $\mathbb{P} = \{\mathbb{P}_h(\cdot|\cdot, \cdot)\}_{h=1}^{H}$ and reward functions $\{r_h(\cdot, \cdot)\}_{h=1}^{H}$). Similar to the bandit case, for each episode $k = 1, \cdots, K$, a single agent $m = m_k$ is chosen to participate. An episode $k$ begins with an initial state $s_1^k$, which is drawn from an unknown fixed distribution. Then for steps $h = 1, \cdots, H$, the participating agent $m$ selects an action $a_h^k$ based on the observed state $s_h^k$. After each action, the agent receives a reward $r_h^k = r_h(s_h^k, a_h^k)$, where $r_h : \mathcal{S} \times \mathcal{A} \to \mathbb{R}$ is the reward function at step $h$. Here for the sake of convenience, we assume the reward function to be deterministic, but it is not difficult to generalize our result to stochastic rewards. We also assume $r_h(s, a) \in [0, 1]$ for all $(s, a) \in \mathcal{S} \times \mathcal{A}$ without loss of generality. The environment then transitions to the next state according to $s_{h+1}^k \sim \mathbb{P}_h(\cdot|s_h^k, a_h^k)$, where $\mathbb{P}_h$ is the transition probability at step $h$. The episode terminates when $r_H$ is observed.

The strategy an agent employs to interact with the environment is called the agent's *policy*, which can be described by a set of decision functions $\pi = \{\pi_h\}_{h=1}^{H}$, where $\pi_h : \mathcal{S} \to \mathcal{A}$ is the decision function at level $h$, mapping the current state to an action to select.

**Value Functions.** For any policy $\pi = \{\pi_h\}$, we define $Q$-value functions and $V$-value functions:

$$Q_h^\pi(s_h, a_h) := \mathbb{E}\bigg[\sum_{h'=h}^{H} r_{h'}(s_{h'}, a_{h'}) \bigg| s_h, a_h\bigg], \quad V_h^\pi(s_h) := \mathbb{E}\bigg[\sum_{h'=h}^{H} r_{h'}(s_{h'}, a_{h'}) \bigg| s_h\bigg], \tag{2}$$

where the expectation is taken over the trajectory $(s_1, a_1, \cdots, s_h, a_h)$, determined by the transition probability functions $\mathbb{P}$ and policy $\pi$. The optimal strategy $\pi^*$ is the maximizer of the value functions:

$$\pi^* := \text{argmax}_\pi V_1^\pi(s_1), \forall s_1.$$

We also have optimal value functions $Q_h^* := Q_h^{\pi^*}$ and $V_h^* := V_h^{\pi^*}$, which satisfy Bellman equations

$$Q_h^*(s_h, a_h) = r_h(s_h, a_h) + \mathbb{E}\big[V_{h+1}^*(s_{h+1}) \big| s_h, a_h\big], \quad V_h^*(s_h) = \max_{a \in \mathcal{A}} Q_h^*(s_h, a). \tag{3}$$

**Function Approximation.** We approximate $Q$-value functions with function classes $\{\mathcal{F}_h\}_{h=1}^{H}$, which contain real value functions with domain $\mathcal{S} \times \mathcal{A}$. One basic assumption is that $Q_h^* \in \mathcal{F}_h$ for all steps $h \in [H]$. Now with the convention that functions at level $H + 1$ are uniformly zero, i.e., $f_{H+1} = 0$, we define the Bellman operator $\mathcal{T}_h$:

$$(\mathcal{T}_h f_{h+1})(s_h, a_h) := \mathbb{E}\big[r_h(s_h, a_h) + f_{h+1}(s_{h+1}) \big| s_h, a_h\big],$$

and we expect $\mathcal{T}_h$ to map any function in $\mathcal{F}_{h+1}$ to a function in $\mathcal{F}_h$, i.e., $\mathcal{T}_h \mathcal{F}_{h+1} \subseteq \mathcal{F}_h$. This is called the completeness assumption, which is a fundamental assumption in RL with general function approximation [Wang et al., 2020b, Jin et al., 2021].

**Learning Objective.** The primary goal in multi-agent MDPs is to minimize the cumulative regret over $K$ episodes

$$\text{Reg}(K) = \sum_{k=1}^{K}\left[V_1^*(s_1^k) - V_1^{\pi_{m,k}}(s_1^k)\right],$$

where $\pi_{m,k}$ is the policy of agent $m = m_k$ at round $k$, while the secondary objective is to minimize the communication cost.

### 3.3 Eluder Dimension and Covering Number

To measure the complexity of the learning objective, Russo and Van Roy [2013] first proposed the concept of Eluder dimension, which we define below.

**Definition 3.2** ($\epsilon$-dependence). For a function class $\mathcal{F}$ on domain $\mathcal{D}$, a point $z \in \mathcal{D}$ is $\epsilon$-*dependent* on $\mathcal{Z} \subseteq \mathcal{D}$ if, for any $f_1, f_2 \in \mathcal{F}$ satisfying $\sqrt{\sum_{z' \in \mathcal{Z}} \left(f_1(z') - f_2(z')\right)^2} \leq \epsilon$, it must hold that $|f_1(z) - f_2(z)| \leq \epsilon$. Accordingly, $z$ is $\epsilon$-*independent* of $\mathcal{Z}$ if it is not $\epsilon$-dependent on $\mathcal{Z}$.

**Definition 3.3** (Eluder dimension). The $\epsilon$-*Eluder dimension* $\dim_E(\mathcal{F}, \epsilon)$ is the length of the longest sequence of elements in $\mathcal{D}$ satisfying that, for some $\epsilon_0 > \epsilon$, each element is $\epsilon_0$-independent of the set consisting of its predecessors.

It has been demonstrated that the Eluder dimension roughly corresponds to regular dimension concepts in linear and quadratic cases [Russo and Van Roy, 2013], and that the Eluder family is strictly larger than the generalized linear class [Li et al., 2022]. Note that our Eluder definition can be applied to either the contextual bandit case with $\mathcal{D} = \mathcal{A}$ or the MDPs case with $\mathcal{D} = \mathcal{S} \times \mathcal{A}$.

We also introduce covering number for function classes [Wainwright, 2019] in the following:

**Definition 3.4** (Covering number). An $\epsilon$-*cover* of $\mathcal{F}$ is any subset $\mathcal{F}_\epsilon \subseteq \mathcal{F}$ such that for any $f \in \mathcal{F}$, there exists $f' \in \mathcal{F}_\epsilon$ that $\|f - f'\|_\infty \leq \epsilon$. The *covering number* of $\mathcal{F}$, denoted by $N(\mathcal{F}, \epsilon)$, is the minimal cardinality of its $\epsilon$-cover.

### 3.4 Communication Protocol

We consider a star-shaped communication model [He et al., 2022, Min et al., 2023], where the agents communicate through a central server to collaborate. To ensure asynchronous communication, we mandate that all communications must be initiated by a participating agent. Specifically, at the end of a time step / episode, the agent will decide whether or not to trigger a communication round. If so, the agent uploads its local data history and receives some global data for future decision making. The *communication cost* is the total number of communication rounds initiated by the agents.

One variability is the form of global data that the communicating agent downloads from server. It may be tempting to have the server send all its stored trajectories to the agent for future decision making, but this will unnecessarily expose other agents' data to the current participating agent. We will come back to this issue and our solution in Section 4.2.

## 4 Multi-Agent Contextual Bandits

In this section, we introduce the Asynchronous Nonlinear UCB (Async-NLin-UCB) algorithm designed for multi-agent contextual bandits with general function approximation, and provide a theoretical result for its regret and communication cost.

### 4.1 Algorithm: Async-NLin-UCB

Algorithm 1 takes as input the total number of time steps $T$, regularization parameter $\lambda$, communication parameter $\alpha$ and exploration radii $\{\beta_t\}_{t=1}^T$.

In the algorithm, there are some variables that go through different versions as $t$ progresses through $1, \cdots, T$. For clarity, here we give them an extra subscript $t$ to denote the version of that variable before (not included) the least squares calculation on Line 12 at round $t$.

Throughout the learning process, the server maintains a global history set $Z_t^{\text{ser}}$ that stores action-reward pairs $(a, r) \in \mathcal{A} \times [0, 1]$, initialized on Line 2 and updated only during communication rounds. Each local agent $m$ maintains a decision function $f_{m,t}$ for taking action, a bonus function $b_{m,t}$ for checking communication criterion, and a local data history set $Z_{m,t}^{\text{loc}}$, all initialized on Line 3. Each step of Algorithm 1 contains two parts: local exploration and server updates.

**Part I: Local Exploration.** At step $t$ a single agent $m = m_t$ is active (Line 5). It receives a decision set, finds the greedy action according to its decision function $f_{m,t}$, receives a reward, and updates its local dataset $Z_{m,t}^{\text{loc}}$ (Lines 5 - 7).

**Algorithm 1** Async-NLin-UCB

1: **Input:** total number of rounds $T$, parameters $\lambda$, $\alpha$, $\beta_t$ for $t = 1, \ldots, T$.
2: **Server init:** Set $Z^{\text{ser}} = \varnothing$.
3: **Local init:** For all $m \in [M]$, set $f_m = 1$, $b_m = \mathcal{B}_{\mathcal{A}}(\varnothing, \mathcal{F}; \lambda, \beta_0)$ and $Z_m^{\text{loc}} = \varnothing$.
4: **for** $t = 1, \ldots, T$ **do**
5:     Agent $m = m_t \in [M]$ is active.
6:     Receive decision set $\mathcal{A}_t \subseteq \mathcal{A}$ and take action $a_t \in \operatorname{argmax}_{a \in D_t} f_m(a)$ and receive reward $r_t$.
7:     Update local history $Z_m^{\text{loc}} = Z_m^{\text{loc}} \cup \{(a_t, r_t)\}$.
8:     **if** switch condition (4) is met **then**
9:         Send new data $Z_m^{\text{loc}}$ to server.
10:        **on server**:
11:            Update $Z^{\text{ser}} = Z^{\text{ser}} \cup Z_m^{\text{loc}}$.
12:            Calculate $\widehat{f}$ according to (5) and the bonus function $b = \mathcal{B}_{\mathcal{A}}(Z^{\text{ser}}, \mathcal{F}; \lambda, \beta_t)$.
13:            Send $\widehat{f} + b$ and $b$ to agent $m$.
14:        **end of server**
15:        Agent $m$ receives decision and bonus functions $f_m = \widehat{f} + b$, $b_m = b$, then set $Z_m^{\text{loc}} = \varnothing$.
16:    **end if**
17: **end for**

After exploration, the agent checks if the switch condition is true using its bonus function:

$$\sum_{(a,r) \in Z_{m,t}^{\text{loc}}} b_{m,t}^2(a) / (\beta_{t'}^2 + \lambda) \geq \alpha, \tag{4}$$

where $t'$ is the last time step when agent $m$ communicated with the server. If so, the agent initiates a communication round and uploads its local data (Line 9), prompting the server to begin global policy updates. We will discuss the reasons behind this switch condition in Section 4.2.

**Part II: Server Updates.** After receiving a new local data history from an agent, the server merges the data into its global dataset $Z_t^{\text{ser}}$ (Line 11), and calculate a function $\widehat{f}_{t+1} \in \mathcal{F}$ which minimizes the sum of squares error according to the current dataset $Z_t^{\text{ser}}$ (Line 12):

$$\widehat{f}_{t+1} = \operatorname{argmin}_{f \in \mathcal{F}} \sum_{(a,r) \in Z_t^{\text{ser}}} \big(f(a) - r\big)^2. \tag{5}$$

The next step is to obtain a bonus function $b_{t+1}$ from the oracle $\mathcal{B}_{\mathcal{A}}$ from Definition 4.1 (Line **??**). We discuss the specifics of this construction in detail up next in Section 4.2. Finally, the server sends the optimistic value function $\widehat{f}_{t+1} + b_{t+1}$ and the bonus function $b_{t+1}$ back to agent $m$ for future exploration and updates; agent $m$ also resets its local data history to an empty set (Lines 13 and 15).

### 4.2 Uncertainty Estimators and Bonus Functions

In this section, we introduce uncertainty estimators and bonus functions, and give a detailed explanation for our communication criterion (4). Most of these apply to the MDPs setting as well.

**Uncertainty Estimators.** First we define the uncertainty estimator of new data $a$ against data history $Z$, which is considered in many works on bandits and RL with general function approximation [Gentile et al., 2022, Agarwal et al., 2023]:

$$D_{\lambda, \mathcal{F}}(a; Z) = \sup_{f_1, f_2 \in \mathcal{F}} |f_1(a) - f_2(a)| \Big/ \sqrt{\lambda + \sum_{(a',r) \in Z} |f_1(a') - f_2(a')|^2}, \tag{6}$$

here $\lambda$ is the regularization parameter, $\mathcal{F}$ is a function class. Intuitively, the uncertainty estimator measures the difference between functions on new data $a$ against the difference on historical data $Z$.

**Switch Condition Based On Uncertainty Estimators.** The determinant-based criterion is a common technique used in contextual bandits and RL with linear function approximation to reduce policy switching or communication cost [Abbasi-Yadkori et al., 2011]. For nonlinear function approximation, one can use uncertainty estimators to formulate a new form of switch condition:

$$\sum_{(a,r) \in Z_t^{\text{new}}} D_{\lambda, \mathcal{F}}^2(a; Z_t^{\text{old}}) \geq \alpha. \tag{7}$$

where we use $Z_t^{\text{new}}$ and $Z_t^{\text{old}}$ to denote newly accumulated data and old historical data. This criterion has a similar function as the determinant-based criterion in linear settings. Parameter $\alpha$ controls communication frequency: smaller $\alpha$ indicates more frequent communication, more accurate decision functions and smaller regret, thus implying a trade-off between regret and communication cost.

**Bonus Function Oracle.** Next, we introduce bonus functions obtained through oracles that approximate the uncertainty estimators.

**Definition 4.1** (Bonus Function Oracle $\mathcal{B}_\mathcal{D}$). Given domain $\mathcal{D}$, the oracle $\mathcal{B}_\mathcal{D}(Z, \mathcal{F}; \lambda, \beta)$ takes the following as inputs: a dataset $Z$ consisting of a series of data points $(z, e)$, where $z \in \mathcal{D}$ and $e$ is some additional data content; function class $\mathcal{F}$ with functions $f : \mathcal{D} \to \mathbb{R}_{\geq 0}$; regularization parameter $\lambda$ and exploration radius $\beta$. It returns a function $b \in \mathcal{W}_\mathcal{D} : \mathcal{D} \to \mathbb{R}_{\geq 0}$ satisfying for any $z \in \mathcal{D}$ that

- $b(z) \geq \max \left\{ \left| f_1(z) - f_2(z) \right| : f_1, f_2 \in \mathcal{F}, \sum_{(z,e) \in Z} \left( f_1(z) - f_2(z) \right)^2 \leq \beta^2 \right\}$;

- $D_{\lambda, \mathcal{F}}(z; Z) \leq b(z) / \sqrt{\beta^2 + \lambda} \leq C_\mathcal{B} D_{\lambda, \mathcal{F}}(z; Z)$,

where $C_\mathcal{B}$ is an absolute constant.

*Remark* 4.2. Similar bonus function oracles have been proposed in previous works (Definition 3 in Agarwal et al. [2023]). The accessibility of these oracles is also supported by previous works that proposed methods to compute bonus functions [Kong et al., 2023, Wang et al., 2020b]. In this definition, we leave the domain and data format to be variable so the oracle can be applied to both contextual bandits and MDPs. For bandits, the domain is $\mathcal{A}$, and the data format has $z = a$ and $e = r$. The first property of the bonus function guarantees the optimism of decision functions $\widehat{f}_{t+1} + b_{t+1}$ (see Lemma 6.1 for MDPs or Lemma A.2 for bandits), while the second property links bonuses to uncertainty estimators.

**Switch Condition Based On Bonus Functions.** If we try to adapt the switch condition (7) in our setting, a local agent will require access to historical data $Z_t^{\text{old}}$ to calculate uncertainty estimators $D_{\lambda, \mathcal{F}}^2(a; Z_t^{\text{old}})$. For multi-agent learning, this dataset consists of the collective data from all agents, and giving local agent access is a clear violation of data privacy. Our solution is to let local agents download bonus functions and set communication criterion to (4), using bonus functions instead of uncertainty estimators.

**Decision Functions Based On Bonus Functions.** Another benefit of introducing the bonus function is evident from our exploration method in line 6. A common practice for nonlinear RL algorithms is to construct *confidence sets* of functions during policy update, and find the optimal function within the confidence sets during exploration [Agarwal et al., 2023, Ye et al., 2023]. However, in a multi-agent setting, this would involve the download of confidence sets, which is impractical due to the complex nature of function classes. With the bonus function, local agents need only download the *decision function* from the server for future exploration, which for contextual bandits is simply $\widehat{f}_{t+1} + b_{t+1}$.

### 4.3 Theoretical Results

Our main results for Algorithm 1 are summarized in the following theorem, which provides a regret upper bound and communication complexity order.

**Theorem 4.3.** *By taking* $\gamma = O(1/T)$, $\beta_t = C_{\beta,1} \big( \sqrt{\lambda} + RC(M, \alpha) \log(3MN(\mathcal{F}, \gamma)/\delta) \big)$ *and* $C(M, \alpha) = \sqrt{1 + M\alpha} \big( \sqrt{1 + M\alpha} + M\sqrt{\alpha} \big)$, *the regret of Algorithm 1 within $T$ rounds is*

$$O\left( \sqrt{T} \widetilde{\beta}_1 \sqrt{(1 + M\alpha) \dim_E} \log(T/\min\{1, \lambda\}) + (1 + M\alpha) \dim_E \log^2(T/\min\{1, \lambda\}) \right),$$

*where we abbreviate* $\dim_E := \dim_E(\mathcal{F}, \lambda/T)$; *the total communication complexity is*

$$O\left( (1 + M\alpha)^2 / \alpha \dim_E \log^2(T/\min\{1, \lambda\}) \right)$$

*Remark* 4.4. When reduced to linear contextual bandits, where $\dim_E(\mathcal{F}, \lambda/T) = \widetilde{O}(d)$ and $\log N(\mathcal{F}, \gamma) = \widetilde{O}(d)$, our result on regret correspond exactly to Theorem 5.1 of He et al. [2022], except for an extra $1 + M\alpha$ term in the communication cost, an unimportant term when taking $\alpha = 1/M^2$ that comes from the complication of communication cost analysis in nonlinear settings.

## 5 Multi-Agent Reinforcement Learning

In this section, we introduce the Asynchronous Nonlinear Least Squares Value Iteration UCB (Async-NLin-UCB) algorithm for multi-agent MDPs with general function approximation, and a corresponding theoretical result.

### 5.1 Algorithm: Async-NLSVI-UCB

To better represent the elements in the datasets, we sometimes use $o_h$ to represent the tuple $(s_h, a_h, r_h, s_{h+1})$ and $z_h$ to represent $(s_h, a_h)$ when there is no confusion. Similar to the bandit case, we give some variables an extra subscript $k$ here for clarity, which denotes the version of the variable before (not included) Line 14 at episode $k$.

---

**Algorithm 2** Federated Nonlinear MDPs

---

1: **Input:** total number of rounds $K$, parameters $\lambda$, $\alpha$, $\beta_{k,h}$ for $k = [K]$ and $h \in [H]$
2: **Server init:** Set $Z_h^{\text{ser}} = \varnothing$ for all $h \in [H]$.
3: **Local init:** $\forall m \in [M]$ and $h \in [H]$, set $Q_{m,h} = 1$, $b_{m,h} = \mathcal{B}(\varnothing, \mathcal{F}_h; \lambda, \beta_{0,h})$, $Z_{m,h}^{\text{loc}} = \varnothing$.
4: **for** $k = 1, \ldots, K$ **do**
5:  Agent $m = m_k \in [M]$ is active and receives initial state $s_1^k \in \mathcal{S}$.
6:  **for** $h = 1, \ldots, H$ **do**
7:   Take action $a_h^k = \operatorname{argmax}_{a \in \mathcal{A}} Q_{m,h}(s_h^k, a)$, receive reward $r_h^k$ and next state $s_{h+1}^k$.
8:   Update $Z_{m,h}^{\text{loc}} = Z_{m,h}^{\text{loc}} \cup \{(s_h^k, a_h^k, r_h^k, s_{h+1}^k)\}$.
9:  **end for**
10:  **if** switch condition (8) is met **then**
11:   Send new data $\{Z_{m,h}^{\text{loc}}\}_{h \in [H]}$ to server.
12:   **on server**:
13:    Update $Z_h^{\text{ser}} = Z_h^{\text{ser}} \cup Z_{m,h}^{\text{loc}}$.
14:    Initialize $Q_{H+1} = V_{H+1} = 0$.
15:    **for** $h = H, H-1, \cdots, 1$ **do**
16:     Calculate $\widehat{f}_h$ according to (9) and bonus function $b_h = \mathcal{B}_{\mathcal{S} \times \mathcal{A}}(Z_h^{\text{ser}}, \mathcal{F}_h; \lambda, \beta_{k,h})$.
17:     Calculate $Q_h$ and $V_h$ according to (11).
18:    **end for**
19:    Send $\{Q_h\}_{h=1}^H$ and $\{b_h\}_{h=1}^H$ to agent $m$.
20:   **end of server**
21:   Agent $m$ receives $Q_{m,h} = Q_h$, $b_{m,h} = b_h$ and resets $Z_{m,h}^{\text{loc}} = \varnothing$ for all $h \in [H]$.
22:  **end if**
23: **end for**

---

The server maintains global historical datasets $Z_{k,h}^{\text{ser}}$ containing sequences of tuples $(s_h, a_h, r_h, s_{h+1})$, initialized in Line 2. Each local agent $m$ maintains optimistic value functions $\{Q_{m,k,h}\}_{h=1}^H$, bonus functions $\{b_{m,k,h}\}_{h=1}^H$, and local datasets $\{Z_{m,k,h}^{\text{loc}}\}_{h=1}^H$, all initialized in Line 3.

Each episode $k$ of Algorithm 2 also consists of the two parts local exploration and server updates.

**Part I: Local Exploration.** At step $k$ an agent $m = m_k$ is active (Line 5). It interacts with the environment by executing the greedy policy according to $\{Q_{m,k,h}\}_{h=1}^H$, obtaining a trajectory $\{(s_h^k, a_h^k, r_h^k, s_{h+1}^k)\}_{h=1}^H$, which is then stored into the local historical datasets $Z_{m,k,h}^{\text{loc}}$ (lines 6 - 9). After exploration, the agent checks for the following switch condition: there exists $h \in [H]$ so that

$$\sum_{o_h \in Z_{m,k,h}^{\text{loc}}} b_{m,k,h}^2(s_h, a_h) / (\beta_{k',h}^2 + \lambda) \geq \alpha, \tag{8}$$

where $k'$ is the last communication round for $m$. If so, the agent triggers communication (Line 11).

**Part II: Server Updates.** After receiving new data, the server merges it with its global datasets $Z_{k,h}^{\text{ser}}$ (Line 13) and calculates value function estimates $\{Q_{k+1,h}\}_{h=1}^H$ and $\{V_{k+1,h}\}_{h=1}^H$ using LSVI. Suppose we already have $Q$- and $V$-value function estimates $Q_{k+1,h+1}$ and $V_{k+1,h+1}$ at level $h + 1$. We solve the least squares problem for $\widehat{f}_h$ to minimize the Bellman error (Line 16):

$$\widehat{f}_{k+1,h} = \operatorname{argmin}_{f_h \in \mathcal{F}_h} \sum_{o_h \in Z_{k,h}^{\text{ser}}} \left( f_h(z_h) - r_h - V_{k+1,h+1}(s_{h+1}) \right)^2. \tag{9}$$

We now also define the uncertainty estimator of a new pair of data $z = (s, a)$ against data history $Z$ with normalization parameter $\lambda$ and function class $\mathcal{F}$ as

$$D_{\lambda, \mathcal{F}}(z; Z) = \sup_{f_1, f_2 \in \mathcal{F}} |f_1(z) - f_2(z)| / \sqrt{\lambda + \sum_{o' \in Z} |f_1(z') - f_2(z')|^2}. \tag{10}$$

Similar to the bandits setting, the uncertainty can be approximated with the bonus function acquired from an oracle $\mathcal{B}_{\mathcal{S} \times \mathcal{A}}$ in Definition 4.1. In this case, the domain $\mathcal{D} = \mathcal{S} \times \mathcal{A}$, and the data format corresponds to $z = (s, a)$ and $e = (r, s')$. Despite these definitions not depending on the step $h$, we expect the parameters $z, Z, \mathcal{F}$ to always come from same step $h$. Finally, we allow the bonus function classes $\mathcal{W}_h = \mathcal{W}_{h, \mathcal{S} \times \mathcal{A}}$ to vary between different levels.

After calling oracle for $b_{k+1,h}$ (Line 16), we can obtain value function estimates (Line 17):

$$Q_{k+1,h}(s, a) = \widehat{f}_{k+1,h}(s, a) + b_{k+1,h}(s, a), \quad V_{k+1,h}(s) = \sup_{a \in \mathcal{A}} Q_{k+1,h}(s, a). \tag{11}$$

Iterating through $h = H, \cdots, 1$, the server calculates a set of updated $Q$-value functions $\{Q_{k+1,h}\}_{h=1}^H$ and bonus functions $\{b_{k+1,h}\}_{h=1}^H$, and send them back to agent $m$ for future exploration and updates (lines 19 and 21).

## 5.2 Theoretical Results

We summarize the regret and communication cost of Algorithm 2 in the following theorem:

**Theorem 5.1.** *Taking* $\gamma = O(1/HK)$, $\beta_{h,k} = C_{\beta,2}\Big[\sqrt{\lambda} + HC(M,\alpha)\sqrt{\log(3HMN(\gamma)/\delta)}\Big]$ *and* $N(\gamma) := \max_h N(\mathcal{F}_h, \gamma)N(\mathcal{F}_{h+1}, \gamma)N(\mathcal{W}_{h+1}, \gamma)$, *the regret within* $K$ *rounds is bounded by*

$$O\Big(H\widetilde{\beta}_2\sqrt{(1+M\alpha)\dim_E K}\log(K/\min\{1,\lambda\}) + H^2(1+M\alpha)\dim_E \log^2(K/\min\{1,\lambda\})\Big).$$

*where we abbreviate* $\dim_E := \dim_E(\mathcal{F}, \lambda/K)$*; the total communication complexity is*

$$O\big(H(1+M\alpha)^2\alpha \dim_E(\mathcal{F}, \lambda/K)\log^2(K/\min\{1,\lambda\})\big).$$

*Remark* 5.2. This result when reduced to linear MDPs correspond well to Theorem 5.1 in Min et al. [2023]. Taking $\alpha = 1/M^2$, we get a regret of $\widetilde{O}\big(H^2\sqrt{K\dim_E \log N} + H^2 \dim_E\big)$ and a communication cost of $\widetilde{O}\big(HM^2 \dim_E\big)$, where $N = \max_h\{N(\mathcal{F}_h, \gamma), N(\mathcal{W}_h, \gamma)\}$.

## 6 Proof Sketch

In this section, we provide an outline for the proof of Theorem 5.1, while a more detailed proof can be found in Appendix B, and the full versions of the following lemmas are in Appendix B.1.

### 6.1 Regret Upper Bound

For the regret upper bound, the first lemma establishes optimism of value function estimates.

**Lemma 6.1.** *Taking* $\beta_{k,h}$ *as in Theorem 5.1, with probability at least* $1-\delta$*, for all* $k$, $z \in \mathcal{S} \times \mathcal{A}$ *and* $h \in [H]$, $|\mathcal{T}_h Q_{k+1,h+1}(z) - \widehat{f}_{k+1,h}(z)| \le b_{k+1,h}(z)$.

This allows us to decompose regret into a sum of bonuses:

$$\text{Reg}(K) = \sum_{k=1}^K \big[V_1^*(s_1^k) - V_1^{\pi_{m,k}}(s_1^k)\big]$$
$$\le \sum_{k=1}^K \sum_{h=1}^H \mathbb{E}_{\pi_{m,k}}\big[Q_{m,k,h} - \mathcal{T}_h Q_{m,k,h+1}\big](s_h^k, a_h^k) \le \sum_{k=1}^K \sum_{h=1}^H 2b_{m,k,h}(s_h^k, a_h^k). \quad (12)$$

The sum of bonuses is equal to the sum of uncertainty up to a constant, which we bound in the following lemma corresponding to the elliptical potential lemma [Abbasi-Yadkori et al., 2011].

**Lemma 6.2.** *Define universal datasets as* $Z_{k,h}^{all} = \{o_h^{k'}\}_{k' \in [k]}$*. Then we have for any* $h \in [H]$:

$$\sum_{k=1}^K D_{\lambda,\mathcal{F}}^2(z_h^k; Z_{k-1,h}^{all}) = O\big(\dim_E(\mathcal{F}, \lambda/K)\log^2(K/\min\{1,\lambda\})\big).$$

Careful examination exposes a problem: the uncertainty $D_{\lambda,\mathcal{F}}(z; Z_{k,h}^{\text{ser}})$ corresponding to bonuses are based on server data $Z_{k,h}^{\text{ser}}$ instead of universal data $Z_{k,h}^{\text{all}}$. The next lemma bridges this gap:

**Lemma 6.3.** *For any* $z \in \mathcal{S} \times \mathcal{A}$, $k \in [K]$, $h \in [H]$, $D_{\lambda,\mathcal{F}}^2(z; Z_{k,h}^{ser}) \le (1+M\alpha)D_{\lambda,\mathcal{F}}^2(z; Z_{k,h}^{all})$.

With these, we can deduce the regret bound from (12).

### 6.2 Communication Cost

For communication cost, we employ an *epoch segmentation* scheme, which defines $N$ epochs segmented by episodes $\{k_i\}_{i=1}^N$, with $k_i$ being the smallest episode satisfying

$$\sum_{o_h \in Z_{k_i,h}^{\text{ser}} \setminus Z_{k_{i-1},h}^{\text{ser}}} \sum_{h=1}^H D_{\lambda,\mathcal{F}_h}^2(z_h; Z_{k_{i-1},h}^{\text{ser}}) \ge 1. \quad (13)$$

This is a generalization of epoch segmentation based on doubling determinants in linear settings, yet the lack of determinant in the nonlinear case dramatically increases its complexity. Intuitively, switch condition (8) suggests an agent must gather a substantial amount of data to trigger communication, yet a careful analysis according to (13) yields a maximum of $M + C/\alpha$ communication rounds within one epoch. With this we only need an upper bound for the number of epochs $N$. This is derived by summing (13) over all epochs, then using Lemma 6.1 and Lemma 6.3 to bound the left hand side.

## 7 Conclusions

We propose the algorithms Async-NLin-UCB and Async-NLSVI-UCB to tackle multi-agent nonlinear contextual bandits and MDPs with asynchronous communication. We prove that our algorithms enjoy low regret and communication cost, which are comparable to previous results.

Our algorithms employ a communication criterion that allows the agents to trigger communication rounds, effectively controlling communication cost while promoting the asynchronous protocol. Moreover, we carefully design the contents of server download to guard against data exposure.

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

## Impact Statement

Our work has the potential to enhance cooperative learning systems across diverse fields. By introducing algorithms that enable efficient collaboration among agents with minimal communication overhead, our research paves the way for advancements in distributed systems, including robotics, traffic management, and distributed sensor networks. This could lead to more adaptive, efficient, and scalable systems capable of tackling complex problems in dynamic environments, ultimately contributing to technological progress and societal well-being.

As far as we can tell, there is hardly any negative social impact from our work, mainly because we do not include experiments apart from our theoretical analysis.

# A The Bandit Case: Proof of Theorem 4.3

Before we begin the analysis of Algorithm 1, we reiterate and add some notations for clarity and convenience. Define the data collected by agent $m$ that has already been uploaded to the server by round $t$ as $Z_{m,t}^{\text{up}}$, and the universal data at round $t$ as $Z_t^{\text{all}}$. Apart from these we also have from the algorithm the datasets $Z_{m,t}^{\text{loc}}$ and $Z_t^{\text{ser}}$. It is not difficult to check that they satisfy the following relation:

$$Z_t^{\text{all}} = \bigcup_{m=1}^{M} \left( Z_{m,t}^{\text{up}} \cup Z_{m,t}^{\text{loc}} \right).$$

Furthermore, when $t$ is not a communication round, we also have

$$Z_t^{\text{ser}} = \bigcup_{m=1}^{M} Z_{m,t}^{\text{up}},$$

and when it is a communication round that

$$Z_t^{\text{ser}} = \left[ \bigcup_{m=1}^{M} Z_{m,t}^{\text{up}} \right] \cup Z_{m_t,t}^{\text{loc}},$$

which will be useful in our proof of Lemma A.1 and B.1 in Section C.1.

Next, we assume that at rounds $0 = t_0 < t_1 < \cdots < t_L < t_{L+1} = T + 1$, the participating agent communicates with the server, where $t_0$ and $t_{K+1}$ are dummy rounds. The subscripts will be denoted as $l = 1, \cdots, L$ in the future.

We now describe a participant reordering trick for our asynchronous multi-agent setting, which we will use multiple times in the proof. The basic idea is that, as long as the *communication order* remains the same, and for any given agent, the *number of rounds* between two consecutive communication rounds remains the same, one can switch the episodes around and change the order of agent participation to a certain degree. For example, we may assume that $m_t = m_{t_l}$ for all $t \in (t_{l-1}, t_l]$ by reordering the participants, which means all participation of any given agent happens immediately *before* a certain communication round; as another example, we may assume $m_t = m_{t_{l-1}}$ for all $t \in [t_{l-1}, t_l)$, which means all participation happen immediately *after* communication rounds. It should be noted that one needs to be careful when utilizing this argument, since switching the participation order changes the values of $t_l$ and many associated elements, so applying this trick twice in succession would lead to contradictions.

For a dataset $Z$, we define the $Z$-norm on function set $\mathcal{F}$ as $\|f\|_Z^2 := \sum_{(a,r) \in Z} f^2(a)$ for any $f \in \mathcal{F}$. Then we have the shortened notation

$$D_{\lambda,\mathcal{F}}(a; Z) = \sup_{f_1, f_2 \in \mathcal{F}} \frac{|f_1(a) - f_2(a)|}{\sqrt{\lambda + \|f_1 - f_2\|_Z^2}}.$$

Finally, we define the confidence set of functions at round $t + 1$ as:

$$\mathcal{F}_{t+1} = \left\{ f \in \mathcal{F} : \sum_{(a,r) \in Z_t^{\text{ser}}} \left( f(a) - \widehat{f}_{t+1}(a) \right)^2 \leq \beta_t^2 \right\}, \tag{14}$$

which is a common construction in reinforcement learning.

## A.1 Auxiliary Lemmas

In this section we present some auxiliary lemmas that will be used in the proof of Theorem 4.3. Note these lemmas correspond well to the lemmas presented in 6, only that these are for the contextual bandit case. The proofs for these lemmas can be found in Section C.

**Lemma A.1.** *For any $t \in [T]$, $m \in [M]$ and $f_1, f_2 \in \mathcal{F}$, as long as agent $m$ does not communicate with the server at time step $t$, we have*

$$\lambda + \sum_{m' \in [M]} \|f_1 - f_2\|_{Z_{m',t}^{\text{up}}}^2 \geq \frac{1}{\alpha} \|f_1 - f_2\|_{Z_{m,t}^{\text{loc}}}^2.$$

*Furthermore, for any $t \in [T]$ and $f_1, f_2 \in \mathcal{F}$,*

$$\lambda + \|f_1 - f_2\|_{Z_t^{\text{ser}}}^2 \geq \frac{1}{1 + M\alpha} \left( \lambda + \|f_1 - f_2\|_{Z_t^{\text{all}}}^2 \right),$$

*and as a corollary, for any $a \in \mathcal{A}$,*

$$D^2_{\lambda,\mathcal{F}}(a; Z^{ser}_t) \leq (1 + M\alpha)D^2_{\lambda,\mathcal{F}}(a; Z^{all}_t)$$

This lemma describes the discrepancy between different datasets. Crucially, it provides a worst case ratio between uncertainty measured on the server dataset and universal dataset. This is an important tool for bridging between the different uncertainty estimators in the following proofs. The proof can be found in Section C.1.

**Lemma A.2.** *By taking $\gamma = O(1/T)$ and*

$$\beta_t = \widetilde{\beta}_1 := C_{\beta,1}\big[\sqrt{\lambda} + \sqrt{(\gamma^2 + \gamma R)T} + RC(M,\alpha)\log(3MN(\mathcal{F},\gamma)/\delta)\big],$$

*with $C_{\beta,1} = 6$, where $C(M,\alpha) := \sqrt{1 + M\alpha} + M\sqrt{\alpha}$, we have $f^* \in \mathcal{F}_{t+1}$ for all $t \in \{t_l\}^L_{l=1}$ with probability at least $1 - \delta$. As a corollary, we also have $|f_*(a) - \widehat{f}_{t+1}(a)| \leq b_{t+1}(a)$ for any $a \in \mathcal{A}_t$ and $t \in \{t_l\}^L_{l=1}$.*

This is the central optimism lemma present in all provably efficient reinforcement learning literature. It states that the confidence function set contains the ground truth function $f^*$ with high probability, and in our case, that the decision function $\widehat{f}_t + b_t$ is optimistic. With this, we define the good event $\mathcal{E}_T = \{f^* \in \mathcal{F}_{t+1}, \forall t \in \{t_l\}^L_{l=1}\}$. Then according to A.2, $\mathbb{P}(\mathcal{E}_T) \geq 1 - \delta$. The proof can be found in Section C.2.

**Lemma A.3.** *The sum of squared uncertainty estimators of new data over all historical data can be bounded as follows with some absolute constant $C_D$:*

$$\sum_{t=1}^{T} D^2_{\lambda,\mathcal{F}}(a_t; Z^{all}_{t-1}) \leq C_D \dim_E(\mathcal{F}, \lambda/T) \log^2(T/\min\{1, \lambda\})$$

This lemma corresponds to the elliptical potential argument from the linear setting [Abbasi-Yadkori et al., 2011]. In the nonlinear setting, this lemma essentially reveals the relationship between the sum of Eluder-like confidence quantities and the Eluder dimension. The proof can be found in Section C.3.

### A.2 The Epoch Segmentation Scheme

In this section, we introduce an epoch segmentation scheme, which is needed for both the regret and communication cost proofs presented in the next two sections. It is a generalization of the epoch segmentation scheme based on doubling determinant in the linear bandits / MDPs setting [He et al., 2022, Min et al., 2023], but the lack of a Gram matrix (used for linear regression) in the nonlinear case complicates matters significantly.

We segment the entire run of $t = 1, \cdots, T$ into $N$ epochs as follows. Define iteratively $0 = l_0 < l_1 < \cdots < l_N \leq L$ as

$$l_i = \min\left\{l > l_{i-1} : \sum_{l'=l_{i-1}+1}^{l} \sum_{(a,r) \in Z^{loc}_{m,t_{l'}}} D^2_{\lambda,\mathcal{F}}(a; Z^{ser}_{t_{l_{i-1}}}) \geq 1\right\},$$

where for a given $l'$ in the summation, $m = m_{t_{l'}}$ is the participating agent at $t_{l'}$. In the iterative process, if the above minimum does not exist, simply define $N = i - 1$ and end the process there. Correspondingly, the $i$-th epoch is defined by the time steps $[t_{l_{i-1}}, t_{l_i})$.

The following sections will make use of this epoch scheme as befit their needs, but here we shall give an upper bound for the total number of epochs $N$. Based on the definition of $l_i$, we have for any

$l_{i-1} \leq l < l_i$ that

$$
\begin{aligned}
1 &\geq \sum_{l'=l_{i-1}+1}^{l} \sum_{(a,r)\in Z_{m_{t_{l'}},t_{l'}}^{\mathrm{loc}}} D_{\lambda,\mathcal{F}}^2(a; Z_{t_{l_{i-1}}}^{\mathrm{ser}}) \\
&= \sum_{l'=l_{i-1}+1}^{l} \sum_{(a,r)\in Z_{t_{l'}}^{\mathrm{ser}}\backslash Z_{t_{l'-1}}^{\mathrm{ser}}} \sup_{f_1,f_2\in\mathcal{F}} \frac{[f_1(a)-f_2(a)]^2}{\lambda + \|f_1-f_2\|_{Z_{t_{l_{i-1}}}^{\mathrm{ser}}}^2} \\
&\geq \sup_{f_1,f_2\in\mathcal{F}} \frac{\sum_{(a,r)\in Z_{t_l}^{\mathrm{ser}}\backslash Z_{t_{l_{i-1}}}^{\mathrm{ser}}} [f_1(a)-f_2(a)]^2}{\lambda + \|f_1-f_2\|_{Z_{t_{l_{i-1}}}^{\mathrm{ser}}}^2} \\
&= \sup_{f_1,f_2\in\mathcal{F}} \frac{\lambda + \|f_1-f_2\|_{Z_{t_l}^{\mathrm{ser}}}^2}{\lambda + \|f_1-f_2\|_{Z_{t_{l_{i-1}}}^{\mathrm{ser}}}^2} - 1,
\end{aligned}
$$

which gives $\lambda + \|f_1-f_2\|_{Z_{t_l}^{\mathrm{ser}}}^2 \leq 2\big(\lambda + \|f_1-f_2\|_{Z_{t_{l_{i-1}}}^{\mathrm{ser}}}^2\big)$ for any $f_1, f_2 \in \mathcal{F}$. Then we have

$$
D_{\lambda,\mathcal{F}}^2(a; Z_{t_{l_{i-1}}}^{\mathrm{ser}}) \leq 2 D_{\lambda,\mathcal{F}}^2(a; Z_{t_l}^{\mathrm{ser}}) \tag{15}
$$

for any $a$, and so

$$
\begin{aligned}
1 &\leq \sum_{(a,r)\in Z_{t_{l_i}}^{\mathrm{ser}}\backslash Z_{t_{l_{i-1}}}^{\mathrm{ser}}} D_{\lambda,\mathcal{F}}^2(a; Z_{t_{l_{i-1}}}^{\mathrm{ser}}) \\
&= \sum_{l=l_{i-1}+1}^{l_i} \sum_{(a,r)\in Z_{t_l}^{\mathrm{ser}}\backslash Z_{t_{l-1}}^{\mathrm{ser}}} D_{\lambda,\mathcal{F}}^2(a; Z_{t_{l_{i-1}}}^{\mathrm{ser}}) \\
&\leq 2 \sum_{l=l_{i-1}+1}^{l_i} \sum_{(a,r)\in Z_{t_l}^{\mathrm{ser}}\backslash Z_{t_{l-1}}^{\mathrm{ser}}} D_{\lambda,\mathcal{F}}^2(a; Z_{t_{l-1}}^{\mathrm{ser}}),
\end{aligned}
$$

and summing over $i = 1, \cdots, N-1$ that:

$$
N - 1 \leq 2 \sum_{l=1}^{L} \sum_{(a,r)\in Z_{t_l}^{\mathrm{ser}}\backslash Z_{t_{l-1}}^{\mathrm{ser}}} D_{\lambda,\mathcal{F}}^2(a; Z_{t_{l-1}}^{\mathrm{ser}}).
$$

If we apply the participant reordering trick and let $m_t = m_{t_l}$ for all $t \in (t_{l-1}, t_l]$ and $l \in [L]$, we get $Z_{t_l}^{\mathrm{ser}}\backslash Z_{t_{l-1}}^{\mathrm{ser}} = \{(a_t, r_t)\}_{t=t_{l-1}+1}^{t_l}$, and so applying Lemma A.1 and Lemma A.3, we get

$$
\begin{aligned}
N - 1 &\leq 2 \sum_{l=1}^{L} \sum_{t=t_{l-1}+1}^{t_l} D_{\lambda,\mathcal{F}}^2(a_t; Z_{t_{l-1}}^{\mathrm{ser}}) \\
&\leq 2(1+M\alpha) \sum_{l=1}^{L} \sum_{t=t_{l-1}+1}^{t_l} D_{\lambda,\mathcal{F}}^2(a_t; Z_{t-1}^{\mathrm{all}}) \\
&\leq 2(1+M\alpha) \sum_{t=1}^{T} D_{\lambda,\mathcal{F}}^2(a_t; Z_{t-1}^{\mathrm{all}}) \\
&\leq C(1+M\alpha) \dim_E(\mathcal{F}, \lambda/T) \log(T/\lambda) \log T,
\end{aligned}
$$

which gives the order of total number of epochs:

$$
N = O\bigg((1+M\alpha) \dim_E(\mathcal{F}, \lambda/T) \log^2(T/\min\{1,\lambda\})\bigg). \tag{16}
$$

Notice that the participant reordering trick is only used to bound the number of epochs, which itself does not depend on the specific order of participation. This is crucial since it suggests this reordering does not change anything essential, and is in fact not necessary for the proof - it just made the proof easier to read. Therefore we can still reorder participants as we see fit in other parts of our proof.

## A.3 Proof of Regret Upper Bound

Now we are ready to prove the first part of Theorem 4.3 concerning the regret upper bound. We begin by applying the participation reordering trick to assume, without loss of generality, that the same agent is active within the rounds $[t_l, t_{l+1} - 1]$, i.e. $m_{t_l} = m_{t_l+1} = \cdots = m_{t_{l+1}-1}$. Under this assumption, we have $t_1 = 1$.

Let $a_t^* := \operatorname{argmax}_{a \in D_t} f_*(a)$ be the best arm at time $t$. Then by Lemma A.2, $f_*(a_t^*) \leq \big(\widehat{f}_{m_t,t} + b_{m_t,t}\big)(a_t^*) \leq \big(\widehat{f}_{m_t,t} + b_{m_t,t}\big)(a_t)$, where the second inequality is due to the choice of $a_t$ at round $t$. Hence we get

$$
\begin{aligned}
\operatorname{Reg}(T) &= \sum_{t=1}^{T} \big[ f_*(a_t^*) - f_*(a_t) \big] \\
&\leq \min\left\{ \sum_{t=1}^{T} \big( \widehat{f}_{m_t,t} + b_{m_t,t} - f^* \big)(a_t), 4 \right\} \\
&\leq \sum_{t=1}^{T} \min\{ 2b_{m_t,t}(a_t), 4 \} \\
&= 2 \sum_{l=1}^{L} \sum_{t=t_l+1}^{t_{l+1}-1} b_{t_l}(a_t) + 2 \sum_{l=1}^{L} \min\{ b_{m_{t_l},t_l}(a_{t_l}), 2 \},
\end{aligned} \tag{17}
$$

where the first inequality is due to $|f| \leq 1$ from Assumption 3.1, and the second inequality again uses Lemma A.2. We first bound the second term here using the epoch scheme in Section A.2. We start by converting the bonus term to uncertainty:

$$
\begin{aligned}
b_{m_{t_l},t_l}(a_{t_l}) &= b_{t_{l-1}}(a_{t_l}) \\
&\leq C_{\mathcal{B}} \sqrt{\beta_{t_l-1}^2 + \lambda} \cdot D_{\lambda,\mathcal{F}}(a_{t_l}; Z_{t_{l-1}}^{\text{ser}}).
\end{aligned} \tag{18}
$$

Now consider the episodes in an epoch $i$, specifically $\{t_{l_{i-1}}, t_{l_{i-1}+1}, \cdots, t_{l_i}\}$. For any $l_{i-1} < l < l_i$, since $Z_{t_{l_{i-1}}}^{\text{ser}} \subseteq Z_{t_{l-1}}^{\text{ser}}$, we can deduce that

$$
D_{\lambda,\mathcal{F}}^2(z_{t_l}; Z_{t_{l-1}}^{\text{ser}}) \leq D_{\lambda,\mathcal{F}}^2(z_{t_l}; Z_{t_{l_{i-1}}}^{\text{ser}}) \leq 2 D_{\lambda,\mathcal{F}}^2(z_{t_l}; Z_{t_l}^{\text{ser}}),
$$

where the second inequality is borrowed from (15) from Section A.2. Therefore continuing from (18),

$$
\begin{aligned}
\sum_{l=1}^{L} \min\{ b_{m_{t_l},t_l}(z_{t_l}), 2 \} &\leq \sum_{l \notin \{l_i\}_{i=1}^N} \left[ \sqrt{2} C_{\mathcal{B}} \sqrt{\beta_{t_l-1}^2 + \lambda} \cdot D_{\lambda,\mathcal{F}}(z_{t_l}; Z_{t_l}^{\text{ser}}) \right] + \sum_{i=1}^{N} 2 \\
&\leq \sqrt{2} C_{\mathcal{B}} \sum_{l=1}^{L} D_{\lambda,\mathcal{F}}(z_{t_l}; Z_{t_l}^{\text{ser}}) \sqrt{\beta_h^2 + \lambda} + 2N.
\end{aligned} \tag{19}
$$

Now combine this result with the first term in (17) and use again (18), we get

$$
\begin{aligned}
\operatorname{Reg}(T) &\leq 2 C_{\mathcal{B}} \sum_{l=1}^{L} \sum_{t=t_l+1}^{t_{l+1}-1} D_{\lambda,\mathcal{F}}(a_t; Z_{t_l}^{\text{ser}}) \sqrt{\beta_{t_l}^2 + \lambda} + 2\sqrt{2} C_{\mathcal{B}} \sum_{l=1}^{L} D_{\lambda,\mathcal{F}}(z_{t_l}; Z_{t_l}^{\text{ser}}) \sqrt{\beta_h^2 + \lambda} + 4N \\
&\leq 2\sqrt{2} C_{\mathcal{B}} \sum_{l=1}^{L} \sum_{t=t_l}^{t_{l+1}-1} D_{\lambda,\mathcal{F}}(a_t; Z_{t_l}^{\text{ser}}) \sqrt{\beta_{t_l}^2 + \lambda} + 4N \\
&\leq 2\sqrt{2} C_{\mathcal{B}} \left[ \sum_{l=1}^{L} \sum_{t=t_l}^{t_{l+1}-1} D_{\lambda,\mathcal{F}}^2(a_t; Z_{t_l}^{\text{ser}}) \right]^{1/2} \left[ \sum_{t=1}^{T} \big( \widetilde{\beta}_1^2 + \lambda \big) \right]^{1/2} + 4N
\end{aligned}
$$

where

$$
\widetilde{\beta}_1 = C_\beta \left[ \sqrt{\lambda} + RC(M, \alpha) \sqrt{\log(3N() M/\delta)} \right].
$$

According to Lemma A.1 and Lemma A.3, the term

$$\sum_{l=1}^{L}\sum_{t=t_l}^{t_{l+1}-1} D^2_{\lambda,\mathcal{F}}(a_t; Z^{\text{ser}}_{t_l}) \leq (1+M\alpha)\sum_{l=1}^{L}\sum_{t=t_l}^{t_{l+1}-1} D^2_{\lambda,\mathcal{F}}(a_t; Z^{\text{all}}_{t-1})$$

$$= (1+M\alpha)\sum_{t=1}^{T} D^2_{\lambda,\mathcal{F}}(a_t; Z^{\text{all}}_{t-1})$$

$$\leq C(1+M\alpha)\dim_E(\mathcal{F},\lambda/T)\log^2\big(T/\min\{1,\lambda\}\big).$$

combining this with (16), we get

$$\text{Reg}(T) \leq C\big[(1+M\alpha)\dim_E(\mathcal{F},\lambda/T)\log^2\big(T/\min\{1,\lambda\}\big)\big]^{1/2}\bigg[\sum_{t=1}^{T}(\beta_t^2+\lambda)\bigg]^{1/2} + 4N$$

$$= O\bigg(\sqrt{T}\widetilde{\beta}_1\sqrt{(1+M\alpha)\dim_E(\mathcal{F},\lambda/T)}\log(T/\min\{1,\lambda\})$$

$$+ (1+M\alpha)\dim_E(\mathcal{F},\lambda/T)\log^2(T/\min\{1,\lambda\})\bigg).$$

## A.4 Proof of Communication Cost

In this section we prove the second part of Theorem 4.3, by calculating the communication complexity. First, for each communication round $t_l$, assume the last time before $t_l$ when the agent $m_{t_l}$ communicated with the server was $t_{l'}$, then

$$\sum_{(a,r)\in Z^{\text{loc}}_{m,t_l}} D^2_{\lambda,\mathcal{F}}(a; Z^{\text{up}}_{m,t_l}) \geq \sum_{(a,r)\in Z^{\text{loc}}_{m,t_l}} \frac{[b_{t_{l'}}(a)/C]^2}{\beta_{t_{l'}}^2+\lambda} \geq \frac{\alpha}{C^2},$$

Now employing the epoch segmentation scheme from section A.2, for the $i$-th epoch consisting of the time steps $[t_{l_{i-1}}, t_{l_i})$, we have the inequality

$$1 \geq \sum_{l=l_{i-1}+1}^{l_i-1}\sum_{(a,r)\in Z^{\text{loc}}_{m,t_l}} D^2_{\lambda,\mathcal{F}}(a; Z^{\text{ser}}_{t_{l_{i-1}}})$$

$$\geq \sum_{l=l_{i-1}+1}^{l_i-1}\sum_{(a,r)\in Z^{\text{loc}}_{m,t_l}} D^2_{\lambda,\mathcal{F}}\big(a; Z^{\text{up}}_{m,t_l} \cup Z^{\text{ser}}_{t_{l_{i-1}}}\big).$$

For $m \in [M]$, assume the agent $m$ communicated with the server a total of $n_m$ times within $[t_{l_{i-1}}, t_{l_i})$. Then except for the first of these communication rounds, for each $l \in [l_{i-1}+1, l_i-1]$ with $m_{t_l} = m$, there exists $l' \in [l_{i-1}, l)$ with $m_{t_{l'}} = m$, thus we have $Z^{\text{up}}_{m,t_l} \supset Z^{\text{up}}_{m,t_{l'}+1} = Z^{\text{ser}}_{t_{l'}} \supset Z^{\text{ser}}_{t_{l_{i-1}}}$. With this we have the corresponding term

$$\sum_{(a,r)\in Z^{\text{loc}}_{m,t_l}} D^2_{\lambda,\mathcal{F}}(a; Z^{\text{up}}_{m,t_l} \cup Z^{\text{ser}}_{t_{l_{i-1}}}) = \sum_{(a,r)\in Z^{\text{loc}}_{m,t_l}} D^2_{\lambda,\mathcal{F}}(a; Z^{\text{up}}_{m,t_l}) \geq \frac{\alpha}{C_{\mathcal{B}}^2},$$

therefore

$$1 \geq \sum_{m=1}^{M}(n_m-1)\cdot\frac{\alpha}{4C^2} \Rightarrow \sum_{m=1}^{M}n_m \leq M + \frac{C_{\mathcal{B}}^2}{\alpha}$$

Notice that $\sum_{m=1}^{M} n_m = l_i - l_{i-1}$ is the number of communication rounds within $[t_{l_{i-1}}, t_{l_i})$, hence summing over $i$ the total number of communication rounds is upper bounded by $N(M + C_{\mathcal{B}}^2/\alpha)$. Combine this with (16), we have the total number of communication rounds throughout the algorithm is

$$O\bigg(\frac{(1+M\alpha)^2}{\alpha}\dim_E(\mathcal{F},\lambda/T)\log^2\big(T/\min\{1,\lambda\}\big)\bigg).$$

## B The MDPs Case: Proof of Theorem 5.1

Similar to the bandit case, we define $Z_{m,k,h}^{\text{loc}}$, $Z_{m,k,h}^{\text{up}}$, $Z_{k,h}^{\text{ser}}$, and $Z_{k,h}^{\text{all}}$ to be the local, uploaded, server and universal data, with corresponding subscripts of agent $m \in [M]$, episode $k \in [K]$, $h \in [H]$.

Suppose at rounds $0 = k_0 < k_1 < \cdots < k_L < k_{L+1} = T + 1$, the participating agent communicates with the server, where $k_0$ and $k_{L+1}$ are dummy rounds.

For a dataset $Z_h$ in the MDPs setting, we again define the $Z_h$-norm on function set $\mathcal{F}_h$ as $\|f\|_Z^2 := \sum_{o_h \in Z} f^2(z_h)$ for any $f \in \mathcal{F}$. As a reminder, the tuples $o_h = (s_h, a_h, r_h, s_{h+1})$ and $z_h = (s_h, a_h)$. Then we have the shortened notation

$$D_{\lambda,\mathcal{F}_h}(z_h; Z_h) = \sup_{f_1,f_2 \in \mathcal{F}_h} \frac{|f_1(z_h) - f_2(z_h)|}{\sqrt{\lambda + \|f_1 - f_2\|_{Z_h}^2}}.$$

Finally, we define the confidence set of functions at round $k + 1$ and step $h$ as:

$$\mathcal{F}_{k+1,h} = \left\{ f \in \mathcal{F}_h : \sum_{o_h \in Z_{k,h}^{\text{ser}}} \left( f(z_h) - \widehat{f}_{k+1,h}(z_h) \right)^2 \le (\beta_{k,h})^2 \right\}. \tag{20}$$

### B.1 Auxiliary Lemmas

In this section we present some auxiliary lemmas that will be used in the proof of Theorem 5.1. These lemmas are generalizations / restatements to the lemmas presented in 6, and their detailed proofs can be found in Section C.

**Lemma B.1** (Restatement of Lemma 6.3). *For any $k \in [K]$, $m \in [M]$, $h \in [H]$ and $f_1, f_2 \in \mathcal{F}$, as long as agent $m$ does not communicate with the server at episode $k$, we have*

$$\lambda + \sum_{m' \in [M]} \|f_1 - f_2\|_{Z_{m',k,h}^{\text{up}}}^2 \ge \frac{1}{\alpha} \|f_1 - f_2\|_{Z_{m,k,h}^{\text{loc}}}^2.$$

*Furthermore, we have for any $k \in [K]$ and $f_1, f_2 \in \mathcal{F}$,*

$$\lambda + \|f_1 - f_2\|_{Z_{k,h}^{\text{ser}}}^2 \ge \frac{1}{1 + M\alpha} \left( \lambda + \|f_1 - f_2\|_{Z_{k,h}^{\text{all}}}^2 \right),$$

*and as a corollary, for any $z = (s, a) \in \mathcal{S} \times \mathcal{A}$,*

$$D_{\lambda,\mathcal{F}}^2(z; Z_{k,h}^{\text{ser}}) \le (1 + M\alpha) D_{\lambda,\mathcal{F}}^2(z; Z_{k,h}^{\text{all}})$$

Similar to Lemma A.1, this lemma provides a worst case ratio between uncertainty measured on the server dataset and universal dataset. The proof can be found in Section C.1.

**Lemma B.2** (Restatement of Lemma 6.1). *By taking $\gamma = 1/(C_\gamma K H)$ with $C_\gamma \ge 20$, as well as*

$$\beta_{k,h} = \widetilde{\beta}_2 := C_{\beta,2} \left[ \sqrt{\lambda} + HC(M, \alpha) \sqrt{\log(3HMN_h(\gamma)/\delta)} \right],$$

*with $C_{\beta,2} = 12$ for all $k \in [K]$ and $h \in [H]$, where $N_h(\gamma) = N(\mathcal{F}_h, \gamma) \cdot N(\mathcal{F}_{h+1}, \gamma) \cdot N(\mathcal{W}_{h+1}, \gamma)$, we have with probability at least $1 - \delta$ that $\mathcal{T}_h Q_{k+1,h+1} \in \mathcal{F}_{k+1,h}$ for all $k \in \{k_l\}_{l=1}^L$ with probability at least $1 - \delta$. As a corollary, we also have $|\mathcal{T}_h Q_{k+1,h+1}(s, a) - \widehat{f}_{k+1,h}(s, a)| \le b_{k+1,h}(s, a)$ for any $(s, a) \in \mathcal{S} \times \mathcal{A}$, $k \in \{k_l\}_{l=1}^L$ and $h \in [H]$.*

This is the central optimism lemma. It states that the Bellman operator of $Q$-value function at level $h + 1$ is within the confidences set at level $h$. The conclusion immediately gives the optimism inequality $\mathcal{T}_h Q_{k+1,h+1}(s, a) \le Q_{k+1,h}(s, a)$, which we will use at the start of the regret upper bound prove. The proof of the lemma can be found in Section C.2.

With this, we define the good event $\mathcal{E}_T = \{\mathcal{T}_h Q_{k+1,h+1} \in \mathcal{F}_{k+1,h}, \forall k \in \{k_l\}_{l=1}^L, h \in [H]\}$. Then according to Lemma B.2, $\mathbb{P}(\mathcal{E}_T) \ge 1 - \delta$.

**Lemma B.3.** *For some absolute constant $C_D$, the following holds for all level $h \in [H]$:*

$$\sum_{k=1}^K D_{\lambda,\mathcal{F}}^2(z_h^k; Z_{k-1,h}^{\text{all}}) \le C_D \dim_E(\mathcal{F}, \lambda/T) \log^2(T/\min\{1, \lambda\})$$

This lemma is essentially the same as Lemma A.3. It reveals the relationship between the sum of Eluder-like confidence quantities and the Eluder dimension. The proof can be found in Section C.3.

## B.2 The Epoch Segmentation Scheme

In this section, we introduce the epoch segmentation scheme for MDPs, which is again needed for both the regret and communication cost proofs presented in the next two sections. All of this is quite similar to the bandit case in Section A.2, but the introduction of multiple levels $h \in [H]$ does complicate things a bit.

We segment the entire run of episodes $k = 1, \cdots, K$ into $N$ epochs as follows. Define iteratively $0 = l_0 < l_1 < \cdots < l_N \le L$ as

$$
l_i = \min \left\{ l > l_{i-1} : \sum_{l'=l_{i-1}+1}^{l} \sum_{h=1}^{H} \sum_{o_h \in Z_{m,k_{l'},h}^{\text{loc}}} D_{\lambda,\mathcal{F}_h}^2(z_h; Z_{k_{l_{i-1}},h}^{\text{ser}}) \ge 1 \right\},
$$

where for a given $l'$ in the summation, $m = m_{k_{l'}}$ is the participating agent at $k_{l'}$. In the iterative process, if the above minimum does not exist, simply define $N = i - 1$ and end the process there. Correspondingly, the $i$-th epoch is defined by the episodes $[k_{l_{i-1}}, k_{l_i})$.

The following sections will make use of this epoch scheme as befit their needs, but here we shall give an upper bound for the total number of epochs $N$. Based on the definition of $l_i$, we have for any $l_{i-1} \le l < l_i$ that

$$
\begin{aligned}
1 &\ge \sum_{l'=l_{i-1}+1}^{l} \sum_{h=1}^{H} \sum_{o_h \in Z_{m,k_{l'},h}^{\text{loc}}} D_{\lambda,\mathcal{F}}^2(z_h; Z_{k_{l_{i-1}},h}^{\text{ser}}) \\
&= \sum_{l'=l_{i-1}+1}^{l} \sum_{h=1}^{H} \sum_{o_h \in Z_{k_{l'},h}^{\text{ser}} \setminus Z_{k_{l'-1},h}^{\text{ser}}} \sup_{f_1,f_2 \in \mathcal{F}} \frac{[f_1(z_h) - f_2(z_h)]^2}{\lambda + \|f_1 - f_2\|_{Z_{k_{l_{i-1}},h}^{\text{ser}}}^2} \\
&\ge \sum_{h=1}^{H} \sup_{f_1,f_2 \in \mathcal{F}_h} \frac{\sum_{o_h \in Z_{k_l,h}^{\text{ser}} \setminus Z_{k_{l_{i-1}},h}^{\text{ser}}} [f_1(z_h) - f_2(z_h)]^2}{\lambda + \|f_1 - f_2\|_{Z_{k_{l_{i-1}},h}^{\text{ser}}}^2} \\
&= \sum_{h=1}^{H} \left[ \sup_{f_1,f_2 \in \mathcal{F}_h} \frac{\lambda + \|f_1 - f_2\|_{Z_{k_l,h}^{\text{ser}}}^2}{\lambda + \|f_1 - f_2\|_{Z_{k_{l_{i-1}},h}^{\text{ser}}}^2} - 1 \right],
\end{aligned}
$$

which gives $\lambda + \|f_1 - f_2\|_{Z_{k_l,h}^{\text{ser}}}^2 \le 2\big(\lambda + \|f_1 - f_2\|_{Z_{k_{l_{i-1}},h}^{\text{ser}}}^2\big)$ for any $h \in [H]$ and $f_1, f_2 \in \mathcal{F}_h$. Then we have

$$
D_{\lambda,\mathcal{F}}^2(z_h; Z_{k_{l_{i-1}},h}^{\text{ser}}) \le 2 D_{\lambda,\mathcal{F}}^2(z_h; Z_{k_l,h}^{\text{ser}}) \tag{21}
$$

for any $h \in [H]$ and $z_h \in \mathcal{S} \times \mathcal{A}$, and so

$$
\begin{aligned}
1 &\le \sum_{l=l_{i-1}+1}^{l_i} \sum_{h=1}^{H} \sum_{o_h \in Z_{m,k_l,h}^{\text{loc}}} D_{\lambda,\mathcal{F}_h}^2(z_h; Z_{k_{l_{i-1}},h}^{\text{ser}}) \\
&\le 2 \sum_{l=l_{i-1}+1}^{l_i} \sum_{h=1}^{H} \sum_{o_h \in Z_{k_l,h}^{\text{ser}} \setminus Z_{k_{l-1},h}^{\text{ser}}} D_{\lambda,\mathcal{F}_h}^2(z_h; Z_{k_{l-1},h}^{\text{ser}}),
\end{aligned}
$$

and summing over $i = 1, \cdots, N - 1$ that:

$$
N - 1 \le 2 \sum_{h=1}^{H} \sum_{l=1}^{L} \sum_{o_h \in Z_{k_l,h}^{\text{ser}} \setminus Z_{k_{l-1},h}^{\text{ser}}} D_{\lambda,\mathcal{F}_h}^2(z_h; Z_{k_{l-1},h}^{\text{ser}}).
$$

If we apply the participant reordering trick and let $m_k = m_{k_l}$ for all $k \in (k_{l-1}, k_l]$ and $l \in [L]$, we get $Z^{\text{ser}}_{k_l,h} \setminus Z^{\text{ser}}_{k_{l-1},h} = \{o^k_h\}^{k_l}_{k=k_{l-1}+1}$, and so applying Lemma 6.3 and Lemma 6.2, we get

$$N - 1 \le 2 \sum_{h=1}^{H} \sum_{l=1}^{L} \sum_{k=k_{l-1}+1}^{k_l} D^2_{\lambda, \mathcal{F}_h}(z^k_h; Z^{\text{ser}}_{k_{l-1},h})$$

$$\le 2(1 + M\alpha) \sum_{h=1}^{H} \sum_{l=1}^{L} \sum_{k=k_{l-1}+1}^{k_l} D^2_{\lambda, \mathcal{F}_h}(z^k_h; Z^{\text{all}}_{k-1,h})$$

$$\le 2(1 + M\alpha) \sum_{h=1}^{H} \sum_{k=1}^{K} D^2_{\lambda, \mathcal{F}_h}(z^k_h; Z^{\text{all}}_{k-1,h})$$

$$\le CH(1 + M\alpha) \dim_E(\mathcal{F}, \lambda/T) \log(T/\lambda) \log T,$$

which gives the order of total number of epochs:

$$N = O\left(H(1 + M\alpha) \dim_E(\mathcal{F}, \lambda/T) \log^2(T/\min\{1, \lambda\})\right). \tag{22}$$

## B.3 Proof of Regret Upper Bound

In this section, we prove the first half of Theorem 5.1, which gives an upper bound for the cumulative regret of Algorithm 2.

Using the participant reordering trick, assume without loss of generality that the same agent is active within the rounds $[k_l, k_{l+1} - 1]$, i.e. $m_{k_l} = m_{k_l+1} = \cdots = m_{k_{l+1}-1}$. Under this assumption, we have $k_1 = 1$.

We first prove via induction that $Q^*_h \le Q_{m,k,h}$ for any $m \in [M]$, $k \in [K]$ and $h \in [H+1]$. This holds true for $h = H + 1$ trivially since both value functions at $H + 1$ are uniformly 0. Suppose we already have $Q^*_{h+1} \le Q_{m,k,h+1}$, we have from Lemma B.2 that for the last communication round $k'$ for agent $m$, the server functions satisfy $\mathcal{T}_h Q_{k'+1,h+1}(s,a) \le \widehat{f}_{k'+1,h}(s,a) + b_{k'+1,h}(s,a) = Q_{k'+1,h}(s,a)$. Couple this with the fact that $Q_{m,k,h} = Q_{k'+1,h}$, we can prove that

$$Q^*_h = \mathcal{T}_h Q^*_{h+1} \le \mathcal{T}_h Q_{m,k,h+1} \le Q_{m,k,h},$$

which finishes the induction process.

Now let $a^{k*}_h := \arg\max_{a \in \mathcal{A}} Q^*_h(s^k_h, a)$ be the best action at time $t$, then $V^*_h(s^k_h) = Q^*_h(s^k_h, a^{k*}_h) \le Q_{m,k,h}(s^k_h, a^{k*}_h) \le Q_{m,k,h}(s^k_h, a^k_h)$, where the second inequality is due to the choice of $a^k_h$ at round $k$. Hence we get

$$\text{Reg}(K) = \sum_{k=1}^{K} \left[ V^*_1(s^k_1) - V^{\pi_k}_1(s^k_1) \right]$$

$$\le \sum_{k=1}^{K} \min \left\{ V_{m,k,1}(s^k_1) - V^{\pi_k}_1(s^k_1), 2H \right\}$$

$$= \sum_{k=1}^{K} \sum_{h=1}^{H} \min \left\{ \mathbb{E}_{\pi_k} \left[ Q_{m,k,h}(s^k_h, a^k_h) - \mathcal{T}_h Q_{m,k,h+1}(s^k_h, a^k_h) \right], 2 \right\}$$

$$= \sum_{k=1}^{K} \sum_{h=1}^{H} \min \left\{ \mathbb{E}_{\pi_k} \left[ \widehat{f}_{k'+1,h}(s^k_h, a^k_h) + b_{k'+1,h}(s^k_h, a^k_h) - \mathcal{T}_h Q_{m,k,h+1}(s^k_h, a^k_h) \right], 2 \right\}$$

$$\le \sum_{k=1}^{K} \sum_{h=1}^{H} \min \left\{ 2b_{k',h}(s^k_h, a^k_h), 2 \right\}$$

$$= 2 \sum_{l=1}^{L} \sum_{k=k_l+1}^{k_{l+1}-1} \sum_{h=1}^{H} b_{k_l+1,h}(z^k_h) + 2 \sum_{l=1}^{L} \sum_{h=1}^{H} \min\{b_{m_{k_l},k_l,h}(z^{k_l}_h), 1\}. \tag{23}$$

where the second equality uses the Value-decomposition Lemma from Jiang et al. [2017], the second inequality uses again Lemma B.2, and from the third inequality onward we let $k'$ be the last time agent $m$ communicated with the server.

We now bound the second term here using the epoch scheme in Section B.2. We start by converting the bonus term to uncertainty:

$$b_{m_{k_l},k_l,h}(z_h^{k_l}) = b_{k_{l-1},h}(z_h^{k_l})$$
$$\leq C_{\mathcal{B}}\sqrt{\beta_{k_l-1,h}^2 + \lambda} \cdot D_{\lambda,\mathcal{F}_h}(z_h^{k_l}; Z_{k_{l-1},h}^{\text{ser}}). \tag{24}$$

Now consider the episodes in an epoch $i$, specifically $\{k_{l_{i-1}}, k_{l_{i-1}+1}, \cdots, k_{l_i}\}$. For any $l_{i-1} < l < l_i$, since $Z_{k_{l_{i-1}},h}^{\text{ser}} \subseteq Z_{k_{l-1},h}^{\text{ser}}$, we can deduce that

$$D_{\lambda,\mathcal{F}_h}^2(z_h^{k_l}; Z_{k_{l-1},h}^{\text{ser}}) \leq D_{\lambda,\mathcal{F}_h}^2(z_h^{k_l}; Z_{k_{l_{i-1}},h}^{\text{ser}}) \leq 2D_{\lambda,\mathcal{F}_h}^2(z_h^{k_l}; Z_{k_l,h}^{\text{ser}}),$$

where the second inequality is borrowed from (21) from Section B.2. Therefore continuing from (24),

$$\sum_{l=1}^{L}\sum_{h=1}^{H}\min\{b_{m_{k_l},k_l,h}(z_h^{k_l}), 1\} \leq \sum_{l\notin\{l_i\}_{i=1}^{N}}\sum_{h=1}^{H}\left[\sqrt{2}C_{\mathcal{B}}\sqrt{\beta_{k_l-1,h}^2 + \lambda} \cdot D_{\lambda,\mathcal{F}_h}(z_h^{k_l}; Z_{k_l,h}^{\text{ser}})\right] + \sum_{i=1}^{N}\sum_{h=1}^{H}1$$
$$\leq \sqrt{2}C_{\mathcal{B}}\sum_{l=1}^{L}\sum_{h=1}^{H}D_{\lambda,\mathcal{F}_h}(z_h^{k_l}; Z_{k_l,h}^{\text{ser}})\sqrt{\beta_h^2 + \lambda} + NH. \tag{25}$$

Now combine this result with the first term in (23) and use again (24), we get

$$\text{Reg}(K) \leq C_{\mathcal{B}}\sum_{l=1}^{L}\sum_{k=k_l+1}^{k_{l+1}-1}\sum_{h=1}^{H}\left[D_{\lambda,\mathcal{F}_h}(z_h^k; Z_{k_l,h}^{\text{ser}})\sqrt{\beta_h^2 + \lambda}\right] + \sqrt{2}C_{\mathcal{B}}\sum_{l=1}^{L}\sum_{h=1}^{H}\left[D_{\lambda,\mathcal{F}_h}(z_h^{k_l}; Z_{k_l,h}^{\text{ser}})\sqrt{\beta_h^2 + \lambda}\right] + NH$$

$$\leq \sqrt{2}C_{\mathcal{B}}\sum_{l=1}^{L}\sum_{k=k_l}^{k_{l+1}-1}\sum_{h=1}^{H}\left[D_{\lambda,\mathcal{F}_h}(z_h^k; Z_{k_l,h}^{\text{ser}})\sqrt{\beta_h^2 + \lambda}\right] + NH$$

$$\leq \sqrt{2}C_{\mathcal{B}}\left[\sum_{l=1}^{L}\sum_{k=k_l}^{k_{l+1}-1}\sum_{h=1}^{H}D_{\lambda,\mathcal{F}_h}^2(z_h^k; Z_{k_l,h}^{\text{ser}})\right]^{1/2}\left[\sum_{k=1}^{K}\sum_{h=1}^{H}(\beta_h^2 + \lambda)\right]^{1/2} + NH.$$

According to Lemma 6.3 and Lemma 6.2, the term

$$\sum_{l=1}^{L}\sum_{k=k_l}^{k_{l+1}-1}\sum_{h=1}^{H}D_{\lambda,\mathcal{F}_h}^2(z_h^k; Z_{k_l,h}^{\text{ser}}) \leq (1+M\alpha)\sum_{l=1}^{L}\sum_{k=k_l}^{k_{l+1}-1}\sum_{h=1}^{H}D_{\lambda,\mathcal{F}_h}^2(z_h^k; Z_{k-1,h}^{\text{all}})$$

$$\leq (1+M\alpha)\sum_{k=1}^{K}\sum_{h=1}^{H}D_{\lambda,\mathcal{F}_h}^2(z_h^k; Z_{k-1,h}^{\text{all}})$$

$$\leq H(1+M\alpha)\dim_E(\mathcal{F}, \lambda/T)\log(T/\lambda)\log T.$$

Now with $\gamma = O(1/KH)$, we have

$$\beta_h = O(1)\beta_{h+1} + C_\beta\left[\sqrt{\lambda} + H\left(\sqrt{(1+M\alpha)\log(3HN_h(\gamma)/\delta)} + M\sqrt{\alpha\log(3HMN_h(\gamma)/\delta)}\right)\right]$$

therefore, with $C(M, \alpha) = \sqrt{1+M\alpha} + M\sqrt{\alpha}$ and the upper bound for number of epochs $N$ in (22), we have

$$\sum_{l=1}^{L}\sum_{k=k_l+1}^{k_{l+1}-1}\sum_{h=1}^{H}b_{k_l,h}(z_h^k)$$

$$\leq O\left(\left[H(1+M\alpha)\dim_E(\mathcal{F}, \lambda/K)\log^2(K/\min\{1,\lambda\})\right]^{1/2}\left[K\sum_{h=1}^{H}(\beta_h^2 + \lambda)\right]^{1/2} + HN\right)$$

$$= O\left(H\sqrt{K}\widetilde{\beta}_2\sqrt{(1+M\alpha)\dim_E(\mathcal{F}, \lambda/K)}\log(K/\min\{1,\lambda\})\right.$$

$$\left. + H^2(1+M\alpha)\dim_E(\mathcal{F}, \lambda/K)\log^2(K/\min\{1,\lambda\})\right),$$

678   where $\widetilde{\beta}_2 = C_{\beta,2}\left[\sqrt{\lambda} + HC(M,\alpha)\log\left(HMN(\mathcal{F},\gamma)N(\mathcal{W},\gamma)/\delta\right)\right]$ is the choice of $\beta_{k,h}$ in the

679   algorithm.

## B.4   Proof of Communication Cost

681   Next up, we calculate the communication complexity of Algorithm 2 and prove the second half of

682   Theorem 5.1. For each communication round $k_l$, assume the last time before $k_l$ when the agent

683   $m = m_{k_l}$ communicated with the server was $k_{l'}$, then by the communication rule there exists

684   $h_l \in [H]$ such that $\sum_{o_{h_l} \in Z^{\text{loc}}_{m,k_l,h_l}} b^2_{k_{l'},h_l}(z_{h_l})/(\beta^2_{k_{l'},h_l} + \lambda) \geq \alpha$,

$$\sum_{o_{h_l} \in Z^{\text{loc}}_{m,k_l,h_l}} D^2_{\lambda,\mathcal{F}_{h_l}}(z_{h_l}; Z^{\text{up}}_{m,k_l,h_l}) \geq \sum_{o_{h_l} \in Z^{\text{loc}}_{m,k_l,h_l}} \frac{\left[b_{k_{l'},h_l}(z_{h_l})/C\right]^2}{\beta^2_{k_{l'},h_l} + \lambda} \geq \frac{\alpha}{C^2},$$

685   Next we will make use of the epoch segmentation scheme in Section B.2. For the $i$-th epoch consisting

686   of the time steps $[k_{l_{i-1}}, k_{l_i})$, we have the inequality

$$1 \geq \sum_{l=l_{i-1}+1}^{l_i-1} \sum_{h=1}^{H} \sum_{o_h \in Z^{\text{loc}}_{m,k_l,h}} D^2_{\lambda,\mathcal{F}_h}(z_h; Z^{\text{ser}}_{k_{l_{i-1}},h})$$

$$\geq \sum_{l=l_{i-1}+1}^{l_i-1} \sum_{h=1}^{H} \sum_{o_h \in Z^{\text{loc}}_{m,k_l,h}} D^2_{\lambda,\mathcal{F}_h}(z_h; Z^{\text{up}}_{m,k_l,h} \cup Z^{\text{ser}}_{k_{l_{i-1}},h}).$$

For $m \in [M]$, assume the agent $m$ communicated with the server a total of $n_m$ times within $[k_{l_{i-1}}, k_{l_i})$. Then except for the first of these communication rounds, for each $l \in [l_{i-1}+1, l_i-1]$ with $m_{k_l} = m$, there exists $l' \in [l_{i-1}, l)$ with $m_{k_{l'}} = m$, thus we have $Z^{\text{up}}_{m,k_l,h} \supset Z^{\text{up}}_{m,k_{l'}+1,h} = Z^{\text{ser}}_{k_{l'},h} \supset Z^{\text{ser}}_{k_{l_{i-1}},h}$ for all $h \in [H]$. With this we have

$$\sum_{h=1}^{H} \sum_{o_h \in Z^{\text{loc}}_{m,k_l,h}} D^2_{\lambda,\mathcal{F}_h}(z_h; Z^{\text{up}}_{m,k_l,h} \cup Z^{\text{ser}}_{k_{l_{i-1}},h}) = \sum_{h=1}^{H} \sum_{o_h \in Z^{\text{loc}}_{m,k_l,h}} D^2_{\lambda,\mathcal{F}_h}(z_h; Z^{\text{up}}_{m,k_l,h}) \geq \frac{\alpha}{4C^2},$$

687   therefore

$$1 \geq \sum_{m=1}^{M}(n_m - 1) \cdot \frac{\alpha}{4C^2} \Rightarrow \sum_{m=1}^{M} n_m \leq M + \frac{4C^2}{\alpha}$$

Notice that $\sum_{m=1}^{M} n_m = l_i - l_{i-1}$ is the number of communication rounds within $[k_{l_{i-1}}, k_{l_i})$, hence summing over $i$ the total number of communication rounds is upper bounded by $N(M + 4C^2/\alpha)$. Combine this with the result in (22), we have the total number of communication rounds throughout the algorithm is

$$O\left(H\frac{(1 + M\alpha)^2}{\alpha}\dim_E(\mathcal{F}, \lambda/K)\log^2(K/\min\{1,\lambda\})\right).$$

## C   Proof of Auxiliary Lemmas

689   In this section we prove all the auxiliary lemmas in Section A.1 and Section $B.1$. Note that some of

690   these lemmas are very similar in nature, for which we will only give the proof for the version for the

691   MDPs case, and briefly remark on the version for the bandit case.

### C.1   Proof of Lemma A.1 and Lemma B.1

693   Here we prove Lemma B.1 in detail. The proof for Lemma A.1 is very similar, and so we will only

694   give a short remark on how to apply this to the bandit case.

695   *Proof of Lemma B.1.* First, for an episode $k \in [K]$ and agent $m \in [M]$ such that $m$ does not

696   communicate with the server at episode $k$ (either $m$ is not participating or $k$ is not a communication

697 round), from the communication criterion we have

$$
\alpha \geq \sum_{o_h \in Z_{m,k,h}^{\mathrm{loc}}} \frac{b_{m,k,h}^2(a)}{\beta_{k',h}^2 + \lambda}
$$

$$
\geq \sum_{o_h \in Z_{m,k,h}^{\mathrm{loc}}} D_{\lambda,\mathcal{F}_h}^2(z_h; Z_{k',h}^{\mathrm{ser}})
$$

$$
= \sum_{o_h \in Z_{m,k,h}^{\mathrm{loc}}} \sup_{f_1,f_2 \in \mathcal{F}_h} \frac{|f_1(z_h) - f_2(z_h)|^2}{\lambda + \|f_1 - f_2\|_{Z_{k',h}^{\mathrm{ser}}}}
$$

$$
\geq \sup_{f_1,f_2 \in \mathcal{F}_h} \frac{\|f_1 - f_2\|_{Z_{m,k,h}^{\mathrm{loc}}}^2}{\lambda + \|f_1 - f_2\|_{Z_{k',h}^{\mathrm{ser}}}^2},
$$

where $k'$ is the last communication round for agent $m$. This means that for any $f_1, f_2 \in \mathcal{F}_h$, $(1/\alpha)\|f_1 - f_2\|_{Z_{m,k,h}^{\mathrm{loc}}}^2 \leq \lambda + \|f_1 - f_2\|_{Z_{k',h}^{\mathrm{ser}}}^2$. Observing that $Z_{k',h}^{\mathrm{ser}} \subset Z_{k,h}^{\mathrm{ser}} = \bigcup_{m'=1}^{M} Z_{m',k,h}^{\mathrm{up}}$ proves the first conclusion that

$$
\frac{1}{\alpha}\|f_1 - f_2\|_{Z_{m,k,h}^{\mathrm{loc}}}^2 \leq \lambda + \sum_{m'=1}^{M} \|f_1 - f_2\|_{Z_{m',k,h}^{\mathrm{up}}}^2.
$$

698 Second, for any $f_1, f_2 \in \mathcal{F}_h$, from the above conclusion we have for any $k \in [K]\backslash\{k_l\}_{l=1}^{L}$ that

$$
\lambda + \|f_1 - f_2\|_{Z_{k,h}^{\mathrm{ser}}}^2 = \lambda + \sum_{m=1}^{M} \|f_1 - f_2\|_{Z_{m,k,h}^{\mathrm{up}}}^2
$$

$$
\geq \frac{1}{M\alpha} \sum_{m=1}^{M} \|f_1 - f_2\|_{Z_{m,k,h}^{\mathrm{loc}}}^2
$$

$$
= \frac{1}{M\alpha} \|f_1 - f_2\|_{Z_{k,h}^{\mathrm{all}} \backslash Z_{k,h}^{\mathrm{ser}}}^2,
$$

699 and when $k = k_l$ for some $l \in [L]$, we have alternatively

$$
\lambda + \|f_1 - f_2\|_{Z_{k,h}^{\mathrm{ser}}}^2 = \lambda + \sum_{m' \neq m_t} \|f_1 - f_2\|_{Z_{m',k,h}^{\mathrm{up}}}^2 + \|f_1 - f_2\|_{Z_{m_k,k,h}^{\mathrm{up}} \cup Z_{m_k,k,h}^{\mathrm{loc}}}^2
$$

$$
\geq \lambda + \sum_{m=1}^{M} \|f_1 - f_2\|_{Z_{m,k,h}^{\mathrm{up}}}^2
$$

$$
\geq \frac{1}{(M-1)\alpha} \sum_{m' \neq m_k} \|f_1 - f_2\|_{Z_{m',k,h}^{\mathrm{loc}}}^2
$$

$$
\geq \frac{1}{M\alpha} \|f_1 - f_2\|_{Z_{k,h}^{\mathrm{all}} \backslash Z_{k,h}^{\mathrm{ser}}}^2.
$$

Either way, we can deduce for any $k \in [K]$ that

$$
(1 + M\alpha)\big(\lambda + \|f_1 - f_2\|_{Z_{k,h}^{\mathrm{ser}}}^2\big) \geq \lambda + \|f_1 - f_2\|_{Z_{k,h}^{\mathrm{all}}}^2.
$$

700 Finally, from the above we immediately have

$$
D_{\lambda,\mathcal{F}}^2(z_h; Z_{k,h}^{\mathrm{ser}}) = \sup_{f_1,f_2 \in \mathcal{F}_h} \frac{[f_1(z_h) - f_2(z_h)]^2}{\lambda + \|f_1 - f_2\|_{Z_{k,h}^{\mathrm{ser}}}^2}
$$

$$
\leq (1 + M\alpha) \sup_{f_1,f_2 \in \mathcal{F}} \frac{[f_1(z_h) - f_2(z_h)]^2}{\lambda + \|f_1 - f_2\|_{Z_{k,h}^{\mathrm{all}}}^2}
$$

$$
= (1 + M\alpha) D_{\lambda,\mathcal{F}}^2(a; Z_{k,h}^{\mathrm{all}}).
$$

701 □

702 *Remark* C.1. Notice that this prove does not depend on the multi-level structure of episodic MDPs,
703 but is a direct result of the communication criterion and protocol. This means the proof can be
704 converted to the bandit case of Lemma A.1 without any essential changes: simply change episode $k$
705 into time step $t$, disregard all mentions of level $h$, and consider $z = a$ instead of $z = (s, a)$.

706 **C.2   Proof of Lemma A.2 and Lemma B.2**

707 We begin with the proof of Lemma A.2, which is an almost direct application of Lemma D.3.

*Proof of Lemma A.2.* We invoke Lemma D.3 with $\epsilon_0 = 0$, then with probability at least $1 - \delta$, for all
$t \in \{t_l\}_{l=1}^L$,

$$\sum_{(a,r)\in Z_t^{\mathrm{ser}}} \left(\widehat{f}_{t+1}(a)-f^*(a)\right)^2 \le C_{\mathrm{ERM}}\left[\lambda+\gamma^2 T+\gamma TR+R^2(1+M\alpha)\log(3N/\delta)+R^2M^2\alpha\log(3NM/\delta)\right] \le \widetilde{\beta}_1^2,$$

708 if we let $\gamma = O(1/T)$ be sufficiently small and take $\widetilde{\beta}_1 = C_{\beta,1}\Big[\sqrt{\lambda} +$

709 $RC(M,\alpha)\log(3MN(\mathcal{F},\gamma)/\delta)\Big]$ with $C_{\beta,1} = \sqrt{C_{\mathrm{ERM}}} = 6$. Thus taking $\beta_t = \widetilde{\beta}_1$, accord-

710 ing to the definition of $\mathcal{F}_{t+1}$, this directly implies $f^* \in \mathcal{F}_{t+1}$.
With this, since the bonus function satisfy

$$b_{t+1}(a) \ge |f_1(a) - f_2(a)|, \quad \forall f_1, f_2 \in \mathcal{F} \quad \text{s.t.} \quad \sum_{(a,r)\in Z_t^{\mathrm{ser}}} \left(f_1(a) - f_2(a)\right)^2 \le \beta_t^2,$$

711 which is based on the first property of the bonus oracle in Definition 4.1, by taking $f_1 = \widehat{f}_{t+1}$ and
712 $f_2 = f^*$ we get for any $a \in \mathcal{A}$ that $b_{t+1}(a) \ge |f_*(a) - \widehat{f}_{t+1}(a)|$, which finishes the proof.   $\square$

713 Next we prove Lemma B.2, which is more challenging and requires an analysis on the least squares
714 value iteration method.

715 *Proof of Lemma B.2.* Take $\mathcal{F}_{h+1,\gamma}$ as a $\gamma$-cover of $\mathcal{F}_{h+1}$, and $\mathcal{W}_{h+1,\gamma}$ as a $\gamma$-cover of $\mathcal{W}_{h+1}$. Select
716 $\bar{f}_{k+1,h+1} \in \mathcal{F}_{h+1,\gamma} \bigoplus \mathcal{W}_{h+1,\gamma}$ so that $\|Q_{k+1,h+1} - \bar{f}_{k+1,h+1}\|_\infty \le \bar{\epsilon} := (1 + \beta_{k+1,h+1})\gamma$. For
717 $o_h = (s_h, a_h, r_h, s_{h+1})$, define the corresponding $y_h = r_h + V_{k+1,h+1}(s_{h+1})$ and $\bar{y}_h = r_h +$
718 $\sup_{a\in\mathcal{A}} \bar{f}_{k+1,h+1}(s_{h+1}, a)$. Let

$$\widetilde{f}_{k+1,h} = \operatorname*{argmin}_{f_h\in\mathcal{F}_h} \sum_{o_h\in Z_{k,h}^{\mathrm{ser}}} \left(f_h(s_h, a_h) - \bar{y}_h\right)^2.$$

719 Then we have

$$\left(\sum_{o_h\in Z_{k,h}^{\mathrm{ser}}} \left(\widehat{f}_{k+1,h}(s_h, a_h) - \bar{y}_h\right)^2\right)^{1/2} \le \left(\sum_{o_h\in Z_{k,h}^{\mathrm{ser}}} \left(\widehat{f}_{k+1,h}(s_h, a_h) - y_h\right)^2\right)^{1/2} + \bar{\epsilon}\sqrt{k}$$

$$\le \left(\sum_{o_h\in Z_{k,h}^{\mathrm{ser}}} \left(\widetilde{f}_{k+1,h}(s_h, a_h) - y_h\right)^2\right)^{1/2} + \bar{\epsilon}\sqrt{k}$$

$$\le \left(\sum_{o_h\in Z_{k,h}^{\mathrm{ser}}} \left(\widetilde{f}_{k+1,h}(s_h, a_h) - \bar{y}_h\right)^2\right)^{1/2} + 2\bar{\epsilon}\sqrt{k}.$$

720 Now notice that $\mathbb{E}\bar{y}_h = \mathcal{T}_h\bar{f}_{k+1,h}(s_h, a_h)$, and the difference $\bar{y}_h - \mathcal{T}_h\bar{f}_{k+1,h}(s_h, a_h)$ is bounded
721 in $[-H, H]$, hence we may apply Lemma D.3 with $f^* = \mathcal{T}_h\bar{f}_{k+1,h}$, $r_t = \bar{y}_h$, $R = H$, $\epsilon_0 = 2\bar{\epsilon}$
722 and $\delta = \delta/3HN(\mathcal{F}_{h+1}, \gamma) \cdot N(\mathcal{W}_{h+1}, \gamma)$, taking a union bound over $\bar{f} \in \mathcal{F}_{h+1,\gamma} \bigoplus \mathcal{W}_{h+1,\gamma}$ and

$h \in [H]$, we have

$$\left( \sum_{o_h \in Z^{\text{ser}}_{k,h}} \left(\widehat{f}_{k+1,h}(s_h,a_h) - \mathcal{T}_h Q_{k+1,h+1}(s_h,a_h)\right)^2 \right)^{1/2}$$

$$\leq \left( \sum_{o_h \in Z^{\text{ser}}_{k,h}} \left(\widehat{f}_{k+1,h}(s_h,a_h) - \mathcal{T}_h \bar{f}_{k+1,h+1}(s_h,a_h)\right)^2 \right)^{1/2} + \gamma\sqrt{k}$$

$$\leq \sqrt{C_{\text{ERM}}}\sqrt{\lambda + (\gamma + 2\bar{\epsilon})^2 K + (\gamma + 2\bar{\epsilon})KH + H^2(1+M\alpha)\log(3HN_h(\gamma)/\delta) + H^2 M^2\alpha\log(3HMN_h(\gamma)/\delta)} + \gamma\sqrt{k}$$

$$\leq \sqrt{C_{\text{ERM}}}\left[ \sqrt{\lambda} + \gamma(3 + 2\beta_{k+1,h+1})\sqrt{K} + \sqrt{\gamma(3 + 2\beta_{k+1,h+1})KH} + HC(M,\alpha)\sqrt{\log(3HMN_h(\gamma)/\delta)} \right],$$

where $N_h(\gamma) = N(\mathcal{F}_h, \gamma) \cdot N(\mathcal{F}_{h+1}, \gamma) \cdot N(\mathcal{W}_{h+1}, \gamma)$. By taking $\gamma = 1/(C_\gamma KH)$ with sufficiently large absolute constant $C_\gamma$ (for example, $C_\gamma = 20$), the second and third terms within the bracket above are both less than $(1/2)\beta_{k+1,h+1}$, and hence we can easily prove via induction on $h$ that the above is no greater than $\widetilde{\beta}_2$, where

$$\widetilde{\beta}_2 = C_{\beta,2}\left[ \sqrt{\lambda} + HC(M,\alpha)\sqrt{\log(3HMN(\gamma)/\delta)} \right]$$

with $C_{\beta,2} = 2\sqrt{C_{\text{ERM}}} = 12$ and $N(\gamma) = \max_{h\in[H]} N_h(\gamma)$. $\qquad\square$

## C.3 Proof of Lemma A.3 and Lemma B.3

In this section we prove Lemma B.3 in detail. The proof for Lemma A.3 is very similar, and so we will again only give a short remark on how to apply this to the bandit case.

*Proof of Lemma B.3.* We fix the level $h \in [H]$ throughout the proof. For an index set $\mathcal{K}_0 \subseteq [K]$, we denote $\mathcal{Z}(\mathcal{K}_0) := \{z^k_h : k \in \mathcal{K}_0\}$.

First, let $n = \lceil \log(K/\lambda)/\log 2 \rceil$, and we divide the set of episodes $\mathcal{K} = [K]$ into $n + 1$ disjoint episode sets as follows. For any $1 \leq l \leq L$ and $k_l \leq k < k_{l+1}$, let

$$(\bar{f}_{k,1}, \bar{f}_{k,2}) = \operatorname*{argmax}_{f_1,f_2 \in \mathcal{F}_h} \frac{\left(f_1(z^k_h) - f_2(z^k_h)\right)^2}{\lambda + \|f_1 - f_2\|^2_{Z^{\text{all}}_{h,k-1}}},$$

and define $L_k : \mathcal{S} \times \mathcal{A} \to \mathbb{R}$ as $L_k(z) = \left(\bar{f}_{k,1}(z) - \bar{f}_{k,2}(z)\right)^2$. Now we define $\mathcal{K}^\iota := \{k \in \mathcal{K} : L_k(z^k_h) \in (2^{-\iota-1}, 2^{-\iota}]\}$ for $\iota \in \{0, 1, \cdots, n-1\}$ and $\mathcal{K}^n := \{k \in \mathcal{K} : L_k(z^k_h) \in [0, 2^{-n}]\}$. We note that for $k \in \mathcal{K}^n$, $L_k(z^k_h) \leq \lambda/K$.

Now define the mapping $\tau : [K] \to [K]$, such that for any $k \in [K]$, $\tau(k)$ is the last episode when agent $m_k$ communicated with the server (not including $k$). We will bound $\sum_{k\in\mathcal{K}^\iota} D^2_{\lambda,\mathcal{F}_h}(z^k_h; Z^{\text{all}}_{h,k-1})$ for $\iota \in \{0, \cdots, n-1\}$.

For a fixed $\iota \leq n-1$, we now decompose $\mathcal{K}^\iota = \bigcup_{j=1}^{n^\iota+1} \mathcal{K}^\iota_j$, where $n^\iota = \lceil |\mathcal{K}^\iota|/\dim_E(\mathcal{F}_h, 2^{-\iota-1}) \rceil$. We start off each set $\mathcal{K}^\iota_j = \varnothing$, and fill them up gradually by iterating through $k \in \mathcal{K}^\iota$ one by one in increasing order to decide which subset $\mathcal{K}^\iota_j$ should $k$ belong to. Specifically, we define $j(k)$ to be the smallest index $j < n^\iota$ such that is $z^k_h$ is $2^{-(\iota+1)/2}$-independent of $\mathcal{Z}(\mathcal{K}^\iota_j)$, and assign $k$ to the set $\mathcal{K}^\iota_{j(k)}$. If such a $j$ does not exist, we simply let $j(k) = n^\iota + 1$ assign $k$ to $\mathcal{K}^\iota_{n^\iota+1}$. Finally after the assignment process, we define $\mathcal{K}^\iota_{j,k} = \mathcal{K}^\iota_j \cap [k]$ for any $k \in [K]$. Then we have the elements added into $\mathcal{K}^\iota_{j(k)-1,k}$ form a sequence where each data corresponding to a new member is $2^{-(\iota+1)/2}$-independent of the old members, and so there are no more than $\dim_E(\mathcal{F}_h, 2^{-\iota-1})$ members within each of them. Moreover, for all $k \in \mathcal{K}^\iota$ that $z^k_h$ is $2^{-(\iota+1)/2}$-dependent on each of $\mathcal{Z}(\mathcal{K}^\iota_{1,k}), \cdots, \mathcal{Z}(\mathcal{K}^\iota_{j(k)-1,k})$.

Now for any $k \in \mathcal{K}^\iota$ by the definition of $\mathcal{K}^\iota$, we have $\left(\bar{f}_{k,1}(z^k_h) - \bar{f}_{k,2}(z^k_h)\right)^2 \geq 2^{-\iota-1}$. This combined with the $2^{-\iota-1}$-dependencies imply that for each $j' = 1, \cdots, j(k)-1$, $\|\bar{f}_{k,1} - \bar{f}_{k,2}\|^2_{\mathcal{Z}(\mathcal{K}^\iota_{j',k})} \geq 2^{-\iota-1}$.

Notice that $\mathcal{Z}(\mathcal{K}^\iota_{j',k}) \subset Z^{\text{all}}_{h,k-1}$ for any $j' \in [j(k)-1]$, and that $\mathcal{Z}(\mathcal{K}^\iota_{j',k})$ for $j' \in [j(k)-1]$ are disjoint, therefore

$$(j(k)-1)2^{-\iota-1} \leq \sum_{j'=1}^{j(k)-1} \|\bar{f}_{k,1} - \bar{f}_{k,2}\|^2_{\mathcal{Z}(\mathcal{K}^\iota_{j',k})} \leq \|\bar{f}_{k,1} - \bar{f}_{k,2}\|^2_{Z^{\text{all}}_{h,k-1}}.$$

It follows that

$$\begin{aligned}
D^2_{\lambda,\mathcal{F}_h}(z^k_h; Z^{\text{all}}_{h,k-1}) &= \frac{\left(\bar{f}_{k,1}(z^k_h) - \bar{f}_{k,2}(z^k_h)\right)^2}{\lambda + \|\bar{f}_{k,1} - \bar{f}_{k,2}\|^2_{Z^{\text{all}}_{h,k-1}}} \\
&\leq \frac{2^{-\iota}}{\lambda + (j(k)-1)2^{-\iota-1}} \\
&= \frac{2}{(j(k)-1) + 2^{\iota+1}\lambda},
\end{aligned}$$

where the first inequality uses the definition of $\mathcal{K}^\iota$. Summing over $k \in \mathcal{K}^\iota$, we have

$$\begin{aligned}
\sum_{k \in \mathcal{K}^\iota} D^2_{\lambda,\mathcal{F}_h}(z^k_h; Z^{\text{all}}_{h,k-1}) &= \sum_{j=1}^{n^\iota+1} \sum_{k \in \mathcal{K}^\iota_j} D^2_{\lambda,\mathcal{F}_h}(z^k_h; Z^{\text{all}}_{h,k-1}) \\
&\leq \sum_{j=1}^{n^\iota} \frac{2|\mathcal{K}^\iota_j|}{(j-1) + 2^{\iota+1}\lambda} + \frac{2|\mathcal{K}^\iota_{n^\iota+1}|}{n^\iota + 2^{\iota+1}\lambda} \\
&\leq \frac{2\dim_E(\mathcal{F}_h, 2^{-\iota-1})}{2^{\iota+1}\lambda} + \sum_{j=2}^{n^\iota} \frac{2\dim_E(\mathcal{F}_h, 2^{-\iota-1})}{j-1} + 2|\mathcal{K}^\iota| \cdot \frac{\dim_E(\mathcal{F}_h, 2^{-\iota-1})}{|\mathcal{K}^\iota|} \\
&\leq \dim_E(\mathcal{F}_h, 2^{-\iota-1})\left(2\log n^\iota + 4 + 1/(2^\iota\lambda)\right),
\end{aligned}$$

where we used the relation $|\mathcal{K}^\iota_j| \leq \dim_E(\mathcal{F}_h, 2^{-\iota-1})$ and the definition of $n^\iota$ in the second inequality. Additionally, for $\iota = n$ we also have

$$\sum_{k \in \mathcal{K}^n} D^2_{\lambda,\mathcal{F}_h}(z^k_h; Z^{\text{all}}_{h,k-1}) \leq \sum_{k \in \mathcal{K}^n} \frac{L_k(z^k_h)}{\lambda} \leq |\mathcal{K}^n| \cdot \frac{\lambda/K}{\lambda} \leq 1,$$

and so finally we sum over $\iota = 0, \cdots, n$ to get

$$\begin{aligned}
\sum_{k=1}^K D^2_{\lambda,\mathcal{F}_h}(z^k_h; Z^{\text{all}}_{h,k-1}) &\leq \sum_{\iota=0}^{n-1} \dim_E(\mathcal{F}_h, 2^{-\iota-1})\left(2\log n^\iota + 4 + 1/(2^\iota\lambda)\right) + 1 \\
&\leq n\dim_E(\mathcal{F}_h, 2^{-n})\left(2\log K + 4 + 1/\lambda\right) + 1 \\
&\leq C\dim_E(\mathcal{F}_h, \lambda/K)\log(K/\min\{1,\lambda\}),
\end{aligned}$$

where the final step makes the assumption that $\lambda = O(1/\log K)$, in which case it holds with some absolute constant $C_D$. $\qquad\square$

*Remark* C.2. Again, this prove does not depend on the multi-level structure of episodic MDPs. In fact, it only relies on the Eluder dimensionality of $\mathcal{F}_h$. This means the proof can be converted to the bandit case of Lemma A.3 without any essential changes: simply change episode $k$ into time step $t$, disregard all mentions of level $h$, and consider $z = a$ instead of $z = (s,a)$.

# D  Technical Lemmas

In this section, we provide a technical concentration lemma that serves as the core of our results. For one, this lemma is based on the following concentration inequality:

**Lemma D.1.** *For a sequence of random variables $\{Z_t\}_{t\in\mathbb{N}}$ adapted to the filtration $\{\mathcal{S}_t\}_{t\in\mathbb{N}}$ and function $f \in \mathcal{F}$, for any $\lambda > 0$, with probability at least $1 - \delta$, for all $t \in \mathbb{N}$, we have*

$$-\frac{1}{\lambda}\sum_{s=1}^t \log\mathbb{E}\left[\exp[-\lambda f(Z_s)]\big|\mathcal{S}_{s-1}\right] - \sum_{s=1}^t f(Z_s) \leq \frac{1}{\lambda\delta}.$$

The proof for this lemma can be found under Lemma 4 of Russo and Van Roy [2013]. Apart from this, we need yet another basic concentration lemma:

**Lemma D.2.** *Suppose $\{\eta_t\}_{t=1}^T$ is a sequence of conditional $R$-sub-Gaussian random variables satisfying $\mathbb{E}\big[e^{\mu \eta_t}\big|\mathcal{H}_{t-1}\big] \leq \exp\big(R^2\mu^2/2\big)$, where $\mathcal{H}_{t-1}$ denotes all history before time $t$, with probability $1 - \delta$, we have*

$$\sum_{t=1}^T \eta_t^2 \leq 2T\sigma^2 + 3\sigma^2 \log(1/\delta).$$

A proof of this lemma can be found under Lemma G.2 of Ye et al. [2023]. With this, we can prove the following lemma characterizing the accuracy of least squares solution. Even though we need this lemma for both bandit and RL settings, we will follow the notations presented in multi-agent contextual bandits. Detailed explanation of how this translates to multi-agent MDPs can be found in Section C.2.

**Lemma D.3.** *Suppose we have a sequence of inputs $\{(a_t, r_t)\}_{t=1}^T$ that follow the rule $r_t = f^*(a_t) + \eta_t$ for some ground truth $f^* \in \mathcal{F}$, with $\eta_t$ being conditionally $R$-sub-Gaussian:*

$$\mathbb{E}\big[e^{\mu \eta_t}\big|a_{1:t}, r_{1:t-1}\big] \leq \exp(R^2\mu^2/2), \forall \mu \in \mathbb{R}.$$

*We also have server datasets $Z_t^{ser}$ at different time steps, collected following the communication protocol in our settings. Note that strictly speaking, the conditions under which $\eta_t$ is sub-Gaussian should also include the former participants $m_{1:t}$, but we will omit this dependency for convenience. Consider $\widehat{f}_{t+1}^{ser}$, the approximate ERM solution to the least squares problem:*

$$\left(\sum_{(a,r)\in Z_t^{ser}} \big(\widehat{f}_{t+1}^{ser}(a) - r\big)^2\right)^{1/2} \leq \min_{f\in\mathcal{F}_t}\left(\sum_{(a,r)\in Z_t^{ser}} \big(f(a) - r\big)^2\right)^{1/2} + \epsilon_0\sqrt{t},$$

*Then abbreviating $N = N(\mathcal{F}, \gamma)$ and taking $C_{ERM} = 36$, with probability at least $1 - \delta$,*

$$\sum_{(a,r)\in Z_t^{ser}} \big(\widehat{f}_{t+1}^{ser}(a) - f^*(a)\big)^2 \leq C_{ERM}\Big[\lambda + (\gamma + \epsilon_0)^2 T + (\gamma + \epsilon_0)TR + R^2(1 + M\alpha)\log(3N/\delta) + R^2M^2\alpha\log(3NM/\delta)\Big]$$

*Proof of Lemma D.3.* Let $\mathcal{F}_\gamma$ be a $\gamma$-cover of the function class $\mathcal{F}$ with respect to the infinity norm $\|\cdot\|_\infty$. For $f \in \mathcal{F}$ and $(a_t, r_t)$ for some $t \in [T]$, let

$$\phi(f, a_t, r_t) = -(f(a_t) - r_t)^2 + (f^*(a_t) - r_t)^2,$$

Since $r_t = f^*(a_t) + \eta_t$, we can write $\phi(f, a_t, r_t)$ as

$$\phi(f, a_t, r_t) = -\big(f(a_t) - f^*(a_t) + \eta_t\big)^2 + \eta_t^2$$
$$= -2\big(f(a_t) - f^*(a_t)\big)\eta_t - \big(f(a_t) - f^*(a_t)\big)^2$$

Since $\eta_t$ is $R$-sub-Gaussian conditional on $Z_{t-1}^{all}, a_t$, we have for any positive parameter $\mu$ that

$$\log\mathbb{E}\big[\exp(\mu\phi(f, a_t, r_t))\big|Z_{t-1}^{all}, a_t\big] \leq 2\mu^2 R^2(f(a_t) - f^*(a_t))^2 - \mu(f(a_t) - f^*(a_t))^2$$
$$= (2\mu^2 R^2 - \mu)(f(a_t) - f^*(a_t))^2$$

Using Lemma D.1, we have with probability at least $1 - \delta/3$, for all $f \in \mathcal{F}_\gamma$ and $t \in [T]$,

$$\mu_{all}\sum_{(a,r)\in Z_t^{all}} \phi(f, a, r) \leq (2\mu_{all}^2 R^2 - \mu_{all})\sum_{(a,r)\in Z_t^{all}} (f(a) - f^*(a))^2 + \log(3N/\delta), \qquad (26)$$

where $\mu_{all} > 0$ is a parameter we will determine later.

On the other hand, if we consider any local agent $m$, when $m_t = m$, we have $\eta_t$ is $R$-sub-Gaussian conditional on $Z_{m,t-1}^{up} \cup Z_{m,t-1}^{loc}$ and $a_t$, i.e. all the data agent $m$ has received from the environment up to this point. Thus we have for any $\mu > 0$ that

$$\log\mathbb{E}\big[\exp(-\mu\phi(f, a_t, r_t))\big|Z_{m,t-1}^{up} \cup Z_{m,t-1}^{loc}, a_t\big] \leq 2\mu^2 R^2(f(a_t) - f^*(a_t))^2 + \mu(f(a_t) - f^*(a_t))^2$$
$$= (2\mu^2 R^2 + \mu)(f(a_t) - f^*(a_t))^2$$

Then again using Lemma D.1 and taking summation on $Z^{\text{loc}}_{m,t}$, with probability at least $1 - \delta/3$, the following holds for any $m \in [M]$:

$$-\mu_{\text{loc}} \sum_{(a,r)\in Z^{\text{loc}}_{m,t}} \phi(f,a,r) \leq (2\mu^2_{\text{loc}}R^2 + \mu_{\text{loc}}) \sum_{(a,r)\in Z^{\text{loc}}_{m,t}} (f(a) - f^*(a))^2 + \log(3NM/\delta), \quad (27)$$

where $\mu_{\text{loc}} > 0$ is a parameter we will determine later.

Taking the summation of (27) for all $m \in [M]$ and combining (26), while observing that $Z^{\text{ser}}_t = Z^{\text{all}}_t \backslash \bigcup_{m=1}^{M} Z^{\text{loc}}_{m,t}$, we get

$$\sum_{(a,r)\in Z^{\text{ser}}_t} \phi(f,a,r) = \sum_{(a,r)\in Z^{\text{all}}_t} \phi(f,a,r) - \sum_{m=1}^{M} \sum_{(a,r)\in Z^{\text{loc}}_{m,t}} \phi(f,a,r)$$

$$\leq (2\mu_{\text{all}}R^2 - 1) \sum_{(a,r)\in Z^{\text{all}}_t} (f(a) - f^*(a))^2 + \frac{1}{\mu_{\text{all}}} \log(3N/\delta)$$

$$+ (2\mu_{\text{loc}}R^2 + 1) \sum_{m=1}^{M} \sum_{(a,r)\in Z^{\text{loc}}_{m,t}} (f(a) - f^*(a))^2 + \frac{1}{\mu_{\text{loc}}} M \log(3NM/\delta)$$

$$= 2R^2(\mu_{\text{all}} + \mu_{\text{loc}})\|f - f^*\|^2_{Z^{\text{all}}_t} - (2\mu_{\text{loc}}R^2 + 1)\|f - f^*\|^2_{Z^{\text{ser}}_t}$$

$$+ \frac{1}{\mu_{\text{all}}} \log(3N/\delta) + \frac{1}{\mu_{\text{loc}}} M \log(3NM/\delta).$$

From Lemma A.1, we have $\lambda + \|f - f^*\|^2_{Z^{\text{all}}_t} \leq (1 + M\alpha)(\lambda + \|f - f^*\|^2_{Z^{\text{ser}}_t}) \Leftrightarrow \|f - f^*\|^2_{Z^{\text{all}}_t} \leq M\alpha\lambda + (1 + M\alpha)\|f - f^*\|^2_{Z^{\text{ser}}_t}$. Plugging this inequality into the above and letting $\mu_{\text{all}} = 1/8R^2(1 + M\alpha)$ and $\mu_{\text{loc}} = 1/8R^2 M\alpha$, we get

$$\sum_{(a,r)\in Z^{\text{ser}}_t} \phi(f,a,r) \leq 2R^2(\mu_{\text{all}} + \mu_{\text{loc}})M\alpha\lambda - (1 - 2M\alpha\mu_{\text{loc}}R^2 - 2(1 + M\alpha)\mu_{\text{all}}R^2)\|f - f^*\|^2_{Z^{\text{ser}}_t}$$

$$+ \frac{1}{\mu_{\text{all}}} \log(3N/\delta) + \frac{1}{\mu_{\text{loc}}} M \log(3NM/\delta)$$

$$\leq -\frac{1}{2}\|f - f^*\|^2_{Z^{\text{ser}}_t} + \frac{1}{2}\lambda + 8R^2(1 + M\alpha)\log(3N/\delta) + 8R^2 M^2 \alpha \log(3NM/\delta). \tag{28}$$

Now for $\widehat{f}^{\text{ser}}_{t+1}$, there exists $\widetilde{f} \in \mathcal{F}_\gamma$ such that $\|\widetilde{f} - \widehat{f}^{\text{ser}}_{t+1}\|_\infty \leq \gamma$. Using Lemma D.2, this gives us the following with probability at least $1 - \delta/3$:

$$-\sum_{(a,r)\in Z^{\text{ser}}_t} \phi(\widetilde{f},a,r) = \sum_{(a,r)\in Z^{\text{ser}}_t} \left[\left(\widetilde{f}(a) - r\right)^2 - \left(f^*(a) - r\right)^2\right]$$

$$\leq \left(\sqrt{\sum_{(a,r)\in Z^{\text{ser}}_t} \left(\widehat{f}^{\text{ser}}_{t+1}(a) - r\right)^2} + \sqrt{t\gamma^2}\right)^2 - \sum_{(a,r)\in Z^{\text{ser}}_t} \left(f^*(a) - r\right)^2$$

$$\leq \left(\sqrt{\sum_{(a,r)\in Z^{\text{ser}}_t} \left(f^*(a) - r\right)^2} + \sqrt{t}(\gamma + \epsilon_0)\right)^2 - \sum_{(a,r)\in Z^{\text{ser}}_t} \left(f^*(a) - r\right)^2$$

$$= (\gamma + \epsilon_0)^2 t + 2(\gamma + \epsilon_0)\sqrt{t}\left(\sum_{s=1}^{t} \eta^2_s\right)^{1/2}$$

$$\leq (\gamma + \epsilon_0)^2 t + 2(\gamma + \epsilon_0)\sqrt{2T^2 R^2 + 3TR^2 \log(3/\delta)},$$

where we used the basic inequality $\sqrt{\sum(a + b)^2} \leq \sqrt{\sum a^2} + \sqrt{\sum b^2}$ in the first inequality and used the property of $\widehat{f}^{\text{ser}}_{t+1}$ in the second inequality. Finally, taking a union bound and combining this

with (28), we have with probability at least $1 - \delta$,

$$\sum_{(a,r) \in Z_t^{\mathrm{ser}}} \left( \widehat{f}_{t+1}^{\mathrm{ser}}(a) - f^*(a) \right)^2$$

$$\leq 2\gamma^2 t + 2 \sum_{(a,r) \in Z_t^{\mathrm{ser}}} \left( \widetilde{f}(a) - f^*(a) \right)^2$$

$$\leq 2\gamma^2 t - 2 \sum_{(a,r) \in Z_t^{\mathrm{ser}}} \phi(\widetilde{f}, a, r) + \lambda + 32R^2(1 + M\alpha) \log(3N/\delta) + 32R^2 M^2 \alpha \log(3NM/\delta)$$

$$\leq 2\gamma^2 T + 2(\gamma + \epsilon_0)^2 T + 4(\gamma + \epsilon_0) \sqrt{2T^2 R^2 + 3TR^2 \log(3/\delta)} + \lambda + 32R^2(1 + M\alpha) \log(3N/\delta) + 32R^2 M^2 \alpha \log(3NM/$$

$$\leq C_{\mathrm{ERM}} \left[ \lambda + (\gamma + \epsilon_0)^2 T + (\gamma + \epsilon_0)TR + R^2(1 + M\alpha) \log(3N/\delta) + R^2 M^2 \alpha \log(3NM/\delta) \right],$$

where the first inequality uses again $\|\widetilde{f} - \widehat{f}_{t+1}^{\mathrm{ser}}\|_\infty \leq \gamma$, and it can be verified that the last inequality holds when $C_{\mathrm{ERM}} \geq 36$. $\qquad \square$

