# OpenReview forum: "Asynchronous Multi-Agent Reinforcement Learning with General Function Approximation"
_NeurIPS.cc/2024/Conference — Submitted to NeurIPS 2024_

### Official Review · Reviewer_39tB · 2024-07-03

**Soundness:** 3
**Presentation:** 3
**Contribution:** 3
**Rating:** 6
**Confidence:** 3

**Summary:**

In this paper, the authors study multi-agent reinforcement learning where agents cooperate through asynchronous communications with a central server to learn a shared environment. They consider the following two settings: multi-agent contextual bandits with general function approximation, and multi-agent RL with general function approximation. For both settings, they propose provably efficient algorithms with low regret and low communication complexity.

**Strengths:**

1. The problem of asynchronous MARL with general function approximation is interesting and important.

2. This paper is the first to consider the setting with general function approximation. The results are solid and the proof looks good to me.

3. For both settings, the authors propose provably efficient algorithms. The results generalize previous results under the linear setting.

**Weaknesses:**

1. It seems that part of the techniques is from previous results, such as the bonus function oracle. It will be helpful if there is a section discussing technical novelty.

2. It seems that the setting is closely related to low switching RL and RL with delayed feedback. It will be interesting if the authors could briefly discuss about the connections.

3. For the communication complexity bound in theorem 5.1, should it be $/\alpha$ instead of $\alpha$? In addition, why not choose $\alpha=1/M$ in both theorems? In this way, the communication cost can be improved. (Please correct me if I misunderstood anything)

4. Line ?? in line 214 of page 6. Please correct it.

**Questions:**

Please refer to the weaknesses above.

**Limitations:**

Yes.

---

> ### Author Rebuttal · Authors · 2024-08-07
>
> We thank the reviewer for the numerous feedback and suggestions to improve our paper. We have corrected the relevant typos and small mistakes, and address the reviewer’s major concerns below:
>
> ---
> **Q1.** It seems that part of the techniques is from previous results, such as the bonus function oracle. It will be helpful if there is a section discussing technical novelty.
>
> **A1.** We highlight some of our main technical novelties in the following. We mentioned the first two of these in the contributions section of our paper (Lines 56-62), while we omitted discussion of the third one due to the intricate technicalities involved and limited space.
>
> - First, our design of the communication content (download data) is quite different from previous works. For multi-agent communication in a linear environment, one may utilize the closed-form solution for least squares regression and transmit the covariance matrix. In contrast, there is no such explicit solution for general function classes, and we designed a data transmission policy consisting of decision functions and bonus functions. This protocol avoids the transmission of the entire dataset, which could result in data leakage. It also does not require the transmission of confidence function sets, which is unrealistic in a nonlinear setting.
>
> - Another key novelty is our communication criterion. The communication criterion in linear settings are formulated by the degree of increment in covariance matrix determinant. Adapting the criterion directly to general function approximation settings is very challenging due to the lack of a closed-form solution to linear regression and a counterpart to covariance matrices.
>
>
>     Our criterion is based on uncertainty estimators, and ensures that our algorithms achieve a regret independent of the number of agents, while maintaining logarithmic communication cost in terms of $T$ and $K$. Intuitively, a communication round is triggered when the local agent collects a substantial amount of new data compared to old data, which implies that the amount of collected data for each communication round grows exponentially.
>
>
>     To the best of our knowledge, the communication criterion we proposed is the first effective criterion for multi-agent nonlinear RL, achieving a regret independent of the number of agents and a communication cost logarithmically dependent on the number of episodes.
>
>
> - One specific technical difficulty is that, when bounding the communication cost, we need to bound the summation of bonus functions across all local data with respect to the previous server update $Z^{\text{ser}}$ (see eq(8)). In contrast, our Lemma 6.2 provides an upper bound for the summation of uncertainty with respect to global data history $Z^{\text{all}}$. The discrepancy between server data $Z^{\text{ser}}$ and global data $Z^{\text{all}}$ presents a unique challenge, and necessitates a meticulous analysis of uncertainty estimators, wherein we have to navigate between these different datasets using the epoch segmentation scheme (see Sections A.2 & B.2).
>
> ---
> **Q2.** It seems that the setting is closely related to low switching RL and RL with delayed feedback. It will be interesting if the authors could briefly discuss about the connections.
>
> **A2.** Thank you for this great suggestion! The design of our communication criterion is indeed partially inspired by the rare-switching strategy in single-agent RL settings, for example eq(3.1) in [1]. Both criteria are used effectively to control the amount of policy updates the agent(s) go through, yet the different problem settings of single-agent versus multi-agent lead to different goals for these criteria. We revised the “Switch Condition Based On Uncertainty Estimators” paragraph (Lines 230-233) to further discuss this:
>
> *This criterion has a similar functionality as the determinant-based criterion in linear settings. It should also be noted that rare switching conditions in single-agent RL with general function approximation have a similar form [1], yet those conditions are used for balancing exploration and exploitation, while our communication criteria are used for balancing regret and communication cost. Specifically, parameter $\alpha$ controls communication frequency: ...*
>
> Apart from this, as we mentioned in the “Switch Condition Based On Bonus Functions” paragraph (Lines 257-262), it is not appropriate to simply use the switch condition used in single-agent settings, as this would grant the local agent access to global datasets, including data collected by other agents. Thus we reformulated the switch condition with bonus functions instead of uncertainty estimators.
>
> On the other hand, we do not see a direct link between our work and RL with delayed feedback. One may consider adapting a similar approach to rare switching to deal with delayed rewards, but our communication criterion is not designed with delayed feedback in mind.
>
> ---
> **Q3.** For the communication complexity bound in theorem 5.1, should it be $/ \alpha$ instead of $\alpha$? In addition, why not choose $\alpha = 1/M$ in both theorems? In this way, the communication cost can be improved.
>
>
> **A3.** For the communication complexity bound in Theorem 5.1, our original version indeed has a typo and the communication cost should have been $O\big( H (1 + M \alpha)^2 / \alpha \dim_E (\mathcal{F}, \lambda / K) \log^2 (K / \min \{1, \lambda\} ) \big)$. As for the value of $\alpha$, notice that $C(M, \alpha) = \sqrt{1 + M\alpha} \big( \sqrt{1 + M\alpha} + M\sqrt{\alpha} \big)$ defined in Theorem 4.3 requires $\alpha = O(1 / M^2)$ to have a constant value independent of $M$, which is a prerequisite for the regret bound to not depend on $M$. We have modified our paper to reiterate the definition of $C(M, \alpha)$ in Theorem 5.1 for clarity.
>
> ---
> [1] Zhao et al. A nearly optimal and low-switching algorithm for reinforcement learning with general function approximation, 2023.

---

> > ### Comment · Reviewer_39tB · 2024-08-13
> >
> > Thanks for your detailed response. I will keep my score.

---

### Official Review · Reviewer_HvUo · 2024-07-10

**Soundness:** 3
**Presentation:** 3
**Contribution:** 3
**Rating:** 7
**Confidence:** 3

**Summary:**

The authors propose two algorithms for asynchronous communication in multi-agent reinforcement learning with generalized value function approximation: Asynchronous-NLin-UCB for context bandit scenarios and Asynchronous-NLSVI-UCB for episodic MDP scenarios. These algorithms achieve near-optimal regret with low communication complexity. The authors theoretically show the trade-off between regret and communication complexity.

**Strengths:**

- The authors provide a detailed background on the related literature concerning regret, communication complexity, and the presence of asynchronous update, which is greatly helpful in understanding the contributions of the proposed algorithms.
- The theoretical foundations and proofs regarding the communication criterion are important and interesting. Also, the trade-off between regret and communication complexity via parameter $\alpha$ offers valuable insights.
- The approach of receiving decisions and bonus functions from a central server instead of historical data is intuitive and appears crucial from a privacy perspective.

**Weaknesses:**

While I have studied general value function approximation, I do not have research background in this field for multi-agent scenarios. Therefore, my critique may not have captured the weaknesses of this paper.
I’m open to revising my score based on the authors' responses.

As far as I know, MARL often adopts the Centralized Training Decentralized Execution(CTDE) framework to avoid the action space growing exponentially with the number of users. However, it is unclear whether the proposed scenario follows "decentralized execution". Agents are supposed to execute based on partial observations in a decentralized manner, but the proposed approach appears to involve a central server consistently during execution. If the proposed scenario is inconsistent with CTDE, I would be interested to hear from the authors what the distinct advantages or necessity of this scenario is.

Typos:

- Line 214: Reference to the label is not correctly written.
- Theorems 4.3 and 5.1: $\tilde{\beta}$ is not properly defined in the statement, and $\beta_t$ should be fixed to $\tilde{\beta}$.
- Theorem 5.1: Total communication complexity should be fixed to $O((1+M\alpha)^2 / \alpha)$.

**Questions:**

A simple question: Why is the order of the Eluder dimension of regret reported as $O(\sqrt{\text{dim}_E})$ instead of $O(\text{dim}_E)$? The regret in Line 677 shows similar results to numerous other papers, but is written in $\sqrt{O(\text{dim}_E)}$ despite the fact that $O(\text{dim}_E)$ dominates. For the linear MDP case, [1] presents a lower bound of $\Omega(dH\sqrt{T})$, and considering $\text{dim}_E = \tilde{O}(d)$, the reported results seem lower than the known lower bound. Many papers report similar results as the authors, but am I missing something?

[1] Zhou Dongruo, Quanquan Gu, and Csaba Szepesvari. "Nearly minimax optimal reinforcement learning for linear mixture Markov decision processes." Conference on Learning Theory. PMLR 2021.

**Limitations:**

The requirement for global state instead of partial observation may limit the practical applicability of the proposed methods.

---

> ### Author Rebuttal · Authors · 2024-08-07
>
> We thank the reviewer for the detailed review and thoughtful questions.  We have fixed all typos mentioned in the review, and address the reviewer’s notable concerns as follows:
>
> ---
> **Q1.** Does the proposed scenario adapt the Centralized Training Decentralized Execution (CTDE) framework in MARL to avoid the action space growing exponentially with the number of users? If not, what are the distinct advantages or necessities of this scenario?
>
> **A1.** Our setting of MARL is different from the CTDE framework. Specifically, in CTDE, agents share the same environment with a joint state, and perform separate actions to maximize a shared reward, while in our setting, the agents operate within their own environments, with separate states, actions and rewards, but try to learn a shared policy by data sharing and collaboration, which is more close to federated reinforcement learning. In this scenario, agents benefit from sharing their learning experiences, which accelerates the learning process. However, designing an efficient communication criterion between different agents becomes a key challenge. We will emphasize this difference in the revision.
>
> ---
> **Q2.** Why is the order of the Eluder dimension of regret reported as $O(\sqrt {\dim_E})$ instead of $O(\dim_E)$, despite the fact that $O(\dim_E)$ dominates? For the linear MDP case, [1] presents a lower bound of $\Omega(dH \sqrt{T})$, and considering $\dim_E \sim \tilde{O} (d)$, the reported results seem lower than the known lower bound.
>
> **A2.** We address the concern regarding the reported results’s dependency on $\dim_E$ in the following two bullet points:
>
> - Discussion about the abbreviated regret bound: The regret bound in Theorem 5.1 has the following format $\tilde{O} (H^2 \sqrt{K \dim_E (\mathcal{F}) \log N (\mathcal{F})} + H^2 \dim_E(\mathcal{F}))$. Despite $\dim_E (\mathcal{F})$ dominating the second term compared to $\sqrt {\dim_E (\mathcal{F})}$ in the first term, the most important term in reinforcement learning is often the one where $K$ dominates. When there is limited space, one typically ignores all other terms and only reports a single term $\tilde{O} (H^2 \sqrt{K \dim_E (\mathcal{F}) \log N (\mathcal{F})}$, making it easier to read and compare to other works.
>
> - Discussion about the lower bound: Compared to previous works such as [2] mentioned by the reviewer, our dependency on dimension $d$ when reduced to the linear setting is also $O(d)$. Apart from $\sqrt{\dim_E (\mathcal{F})}$ contributing a factor of $\sqrt{d}$, $\sqrt{\log N (\mathcal{F})}$ also contributes a factor of $\sqrt{d}$, since the covering number of the linear function space $\mathcal{F}$ is typically exponential in terms of dimension $d$.
>
> ---
>
> [1] Zhou et al. Nearly minimax optimal reinforcement learning for linear mixture Markov decision processes. Proceedings of Thirty Fourth Conference on Learning Theory, 2021
>
> [2] Zhang et al. Multi-Agent Reinforcement Learning: A Selective Overview of Theories and Algorithms, pages 321–384. Springer International Publishing, Cham, 2021. ISBN 978-3-030-60990-0. doi: 10.1007/978-3-030-60990-0_12

---

> ### Comment · Reviewer_HvUo · 2024-08-11
>
> At this point, the author's answers seem to have resolved many of the questions I initially had, and I have a better understanding of their contribution to this field. However, I am still concerned that the paper is simply an extension of linear MDPs to general value function approximation.
> Can the authors explain what new problems they encountered and solved in this extension?

---

> > ### Author Response · Authors · 2024-08-12
> > **Response to Comment**
> >
> > Thank you for the response! We’re glad to hear that we have resolved many of your questions. Regarding the contribution of our work compared with multi-agent RL in the linear MDP setting, we outline some of the main challenges and technical innovations below. The first two challenges are mentioned in the contributions section of our paper (Lines 56-62), but we did not discuss the third one due to its complex technical nature and limited space.
> >
> > - First, our approach to designing the **communication content** (download data) differs significantly from previous work in linear MDPs. For multi-agent communication in a linear environment, one can utilize the closed-form solution for least squares regression and transmit the covariance matrix. However, for general function classes, no explicit solution exists. Thus, we developed a data transmission policy consisting of decision functions and bonus functions. This protocol avoids transmitting the entire dataset, which could result in data leakage, and does not require the sending confidence function sets, which is impractical in a nonlinear context.
> >
> >
> > - Another major innovation is our **communication criterion**. The communication criterion in linear MDPs is based on the degree of increment in covariance matrix determinant. Adapting the criterion directly to general function approximation settings is challenging due to the absence of a closed-form solution to linear regression and a counterpart to covariance matrices.
> >
> >
> >     Our criterion relies on uncertainty estimators. Intuitively, a communication round is triggered when the local agent collects a substantial amount of new data compared to old data, implying that the data collected for each communication round grows exponentially, and thus communication frequency decays exponentially as well. To the best of our knowledge, the communication criterion we proposed is the first effective criterion for multi-agent nonlinear RL, achieving a regret logarithmically dependent on the number of agents while also maintaining logarithmic communication cost with respect to $T$ and $K$.
> >
> >
> > - One particular technical difficulty arises when bounding the communication cost, as we need to bound the summation of bonus functions across all local data concerning the previous server update $Z^{\text{ser}}$ (see eq(8)). In particular, our Lemma 6.2 provides an upper bound for the summation of uncertainty concerning global data history $Z^{\text{all}}$. The **discrepancy between server data $Z^{\text{ser}}$ and global data $Z^{\text{all}}$** presents a unique challenge, requiring careful analysis of uncertainty estimators. This involves navigating between these different datasets using the epoch segmentation scheme (see Appendix A.2 & B.2). This issue also arises in multi-agent linear MDP settings, but it is much simpler to address when uncertainties are measured by matrices.

---

> > > ### Comment · Reviewer_HvUo · 2024-08-13
> > >
> > > Thanks to the reviewers once again for their detailed response. I believe this paper makes a significant contribution to the analysis of federated RL in GVFA, and I will raise my score to 7.

---

### Official Review · Reviewer_ijz1 · 2024-07-11

**Soundness:** 3
**Presentation:** 3
**Contribution:** 2
**Rating:** 5
**Confidence:** 3

**Summary:**

This paper studies the asynchronous multi-agent bandit and RL problem with general function approximation (measured by Elude dimension). The main contribution is to establish $\tilde{O}(\sqrt{\text{dim} T})$ regret bound with $\tilde{O}(M^2 \text{dim})$ communication complexity.

**Strengths:**

This paper is well written and the contribution is solid.

**Weaknesses:**

I think the major concern is non-optimal complexity bounds. Although it seems unreasonable to ask for a matching upper\&lower regret bound for the contextual bandit problem, the part about RL could be possibly improved (at least, the dependence on $H$ is not tight). Also I am curious that what is the current best lower bound for the communication cost to reach an $\sqrt{T}$ regret bound. It would be an interesting problem to study the exact trade-off between the communication cost and regret.

A minor concern might be about the technical novelty given previous methods on measuring the uncertainty.

**Questions:**

What is the meaning of "fully asynchronous" in table 1?

**Limitations:**

Yes.

---

> ### Author Rebuttal · Authors · 2024-08-07
>
> We thank the reviewer for the affirmative review, and address their concerns in the following:
>
>  ---
> **Q1.** The dependency of regret upper bound on $H$ is non-optimal.  Also, what is the current best lower bound for the communication cost to reach an $T$ regret bound?
>
> **A1.** Regarding the optimality of our bounds, we first list and comment on some important past works as reference:
>
> - For the communication complexity, [1] studied the **linear multi-agent MDPs** and propose a lower bound in Theorem 5.5, which we paraphrase below:
>
>     *Given a communication complexity of less than $O(dM)$, the expected regret of an algorithm is at least $\Omega (H \sqrt{dMK})$.*
>
> Since **linear multi-agent MDPs** is included in our general function class with eluder dimension d, it directly implies a $O(dM)$ lower bound in our setting.
>     Compared to our result, this communication complexity corresponds to the case where $\alpha = \Omega(1 / M)$, under which a regret of $O(H^2 \sqrt{dMK})$ is yielded. This is indeed optimal in all parameters except for the number of levels $H$, which we address further below.
>
> - The current optimal regret guarantee for learning **single-agent nonlinear MDPs** is presented in Theorem 4.1 of [2], which achieves a regret upper bound of $\sqrt{HK \dim_{E} \log N}$ for large values of $K$. In comparison, our result is optimal in all parameters except for $H$. The **variance estimator** technique used to remove the extra dependency of $H$ could be also applied to our work and potentially enhance our results. However, since our primary focus is on proposing an algorithm for the multi-agent setting, we chose not to incorporate this technique and suggest it as a promising direction for future research.
>
>
> ---
> **Q2.** A minor concern is the technical novelty given previous methods on measuring the uncertainty.
>
> **A2.** We list some of our main technical novelties in the following, the first two of which we mentioned in the contributions section of our paper (Lines 56-62):
>
> - First, our design of the communication content (download data) is quite different from previous works. For multi-agent communication in a linear environment, one may utilize the closed-form solution for least squares regression and transmit the covariance matrix. In contrast, there is no such explicit solution for general function classes, and we designed a data transmission policy consisting of decision functions and bonus functions. This protocol avoids the transmission of the entire dataset, which could result in data leakage. It also does not require the transmission of confidence function sets, which is unrealistic in a nonlinear setting.
>
> - Another key novelty is our communication criterion. The communication criterion in linear settings are formulated by the degree of increment in covariance matrix determinant. Adapting the criterion directly to general function approximation settings is very challenging due to the lack of a closed-form solution to linear regression and a counterpart to covariance matrices.
>
>
>     Our criterion is based on uncertainty estimators, and ensures that our algorithms achieve a regret independent of the number of agents, while maintaining logarithmic communication cost in terms of $T$ and $K$. Intuitively, a communication round is triggered when the local agent collects a substantial amount of new data compared to old data, which implies that the amount of collected data for each communication round grows exponentially.
>
>
>     To the best of our knowledge, the communication criterion we proposed is the first effective criterion for multi-agent nonlinear RL, achieving a regret independent of the number of agents and a communication cost logarithmically dependent on the number of episodes.
>
>
> - One specific technical difficulty is that, when bounding the communication cost, we need to bound the summation of bonus functions across all local data with respect to the previous server update $Z^{\text{ser}}$ (see eq(8)). In contrast, our Lemma 6.2 provides an upper bound for the summation of uncertainty with respect to global data history $Z^{\text{all}}$. The discrepancy between server data $Z^{\text{ser}}$ and global data $Z^{\text{all}}$ presents a unique challenge, and necessitates a meticulous analysis of uncertainty estimators, wherein we have to navigate between these different datasets using the epoch segmentation scheme (see Sections A.2 & B.2).
>
> ---
> **Q3.** What is the meaning of "fully asynchronous" in table 1?
>
> **A3.** We mainly use the phrase “fully asynchronous” to contrast the setting in [3], where at each round an agent is **chosen to participate** based on a fixed distribution over all agents, and after each communication round the policy update is **sent to all agents**. While this achieves asynchronicity to a certain degree, it does not fully reflect reality where the participation of agents can be arbitrary and completely order-less.  The settings considered in the papers we marked as “fully asynchronous”, on the other hand, allow agents to individually decide when to activate and when to send their history data to the server and request policy updates. In the revision, we clarify this by modifying Lines 83-84 of our paper to the following:
>
> *He et al. [2022] improved communication to be fully asynchronous, where each agent individually and independently interacts with the environment, and proposes the algorithm FedLinUCB with near-optimal regret...*
>
> We also added the same clarification under the table.
>
> ---
> [1] Min et al. Cooperative Multi-Agent Reinforcement Learning: Asynchronous Communication and Linear Function Approximation.
>
> [2] Zhao et al. A nearly optimal and low-switching algorithm for reinforcement learning with general function approximation, 2023.
>
> [3] Li and Wang. Asynchronous upper confidence bound algorithms for federated linear bandits.

---

> > ### Comment · Reviewer_ijz1 · 2024-08-09
> >
> > Thanks for the response. I will adjust my score after discussion with AC and other reviewers.

---

### Official Review · Reviewer_56ui · 2024-07-13

**Soundness:** 2
**Presentation:** 2
**Contribution:** 2
**Rating:** 5
**Confidence:** 4

**Summary:**

This paper studied the distributed federated contextual bandit and federated reinforcement learning (FRL) in the presence of a trusted server. In both problems, nonlinearity and asynchronous communications are explored. Similar algorithms for contextual bandit and FRL that encourage exploration via bonus functions are proposed. Finite-time convergence results in terms of regrets are established for both algorithms and communication complexities are also characterized.

**Strengths:**

* This paper studied the asynchronous federated learning problem where only one agent is activated to sample data and infrequently communicate with server.
* The trigger-based communication is an interesting approach in multi-agent or multi-learner problems.

**Weaknesses:**

* The clarity of some of the important quantities are not well defined or explained. For example,
1)	In the sample complexity result of Theorem 4.3, $\tilde{\beta}_1$ is used. However, it was not defined. It’s unclear what this notation is referring to. Similarly, in Theorem 5.1, $\tilde{\beta}_2$ is used.
2)	The oracle for to compute bonus term bk+1,h is crucial in understanding the algorithms. However, it was not very well-explained or shown anywhere in the main paper.
3)	The two sentences from Line 300 to Line 302 are confusing. Please clarify them.
* Typos:
1)	An extra closing parenthesis appeared in Line 141.
2)	Line 214, ?? -> 12
3)	In Line 1 of algorithm 3, $k=[K]$ -> $k\in [K]$.
4)	In Line 154, the trajectory should be $(s_h, a_h, \cdots, s_H, a_H)$.

**Questions:**

* What if there are more than one agents are to be activated instead of just having one agent at each time t?
* How is an agent chosen to be activated in Line 5 of both Algorithm 1 and 2?
* What is the oracle to compute the bonus term?
* What are the terms $\tilde{\beta}_1$ and $\tilde{\beta}_2$?
* Please clarify the two sentences from Line 300 to Line 302.

**Limitations:**

Please see weaknesses and questions.

---

> ### Author Rebuttal · Authors · 2024-08-07
>
> We thank the reviewer for reading our submission in great detail and pointing out errors and shortcomings. We have fixed all the mentioned typos, and we address other notable concerns in detail below:
>
> ---
> **Q1.** What are the definitions of $\tilde {\beta}_1$ in Theorem 4.3 and $\tilde {\beta}_2$ in Theorem 5.1?
>
> **A1.** We apologize for not including them in our paper.  We have modified the first section of our Theorem 4.3 to the following:
>
> *By taking $\gamma = O (1 / T)$, $\beta_t = \tilde {\beta}_1 = C\_{\beta, 1} \big( \sqrt{\lambda} + R C(M, \alpha) \log (3M N(\mathcal{F}, \gamma) / \delta) \big) $*.
>
> We have also made similar changes to Theorem 5.1.
>
> ---
> **Q2.** The bonus oracle to compute the bonus terms $b_{k + 1, h}$ are not very well-explained or shown in the paper.
>
> **A2.** We defined our bonus oracle $\mathcal{B}$ in Definition 4.1 of our main paper for both bandit and RL problems, under the multi-agent bandit section.  Due to the limited page requirements, we did not redefine the oracle for the MDP case, but instead chose to define the oracle in a more general setting and use the same definition for both cases.  To accommodate, we referred the reader back to Definition 4.1 on line 299 when discussing our algorithm for MDPs.
>
> For the validity of assuming such an oracle exists, many previous works have assumed a similar oracle [1, 2], and other works have justified the use of such an oracle by proposing computation methods for bonus functions [3, 4], as we mentioned in Remark 4.2 after Definition 4.1.  Since the focus of our paper is tackling the asynchronous multi-agent setting, we saw fit not to include too much detail regarding the oracle, serving as a bonus function approximator.
>
> Nevertheless, we thank the reviewer for bringing this issue to our attention, and we will further revise our final paper to ensure the reader is not confused about our oracle definition.
>
> ---
> **Q3.** Please clarify the two sentences on Lines 300-302.
>
> **A3.** We have rewritten our RL algorithm to define different oracles $\mathcal{B}^h \_{\mathcal{S} \times \mathcal{A}}$ for each level $h \in [H]$ for clarity. Correspondingly, we have rewritten our paragraph on oracle explanation (Lines 298-302) to the following:
>
> *Similar to the bandits setting, the uncertainty can be approximated with the bonus function acquired from an oracle $\mathcal{B}^h \_{\mathcal{S} \times \mathcal{A}}$, which is introduced in Definition 4.1. We specify the elements in the definition under MDPs as follows: the domain $\mathcal{D} = \mathcal{S} \times \mathcal{A}$, and the data format corresponds to $z = (s, a)$ and $e = (r, s')$.  Notice that when we call the oracle on Line 16 of Algorithm 2, the elements $Z_h^{\text{ser}}, \mathcal{F}_h$ and the expected return $b_h$ all operate on the same level $h$, therefore we can assume a different oracle for each level $h \in [H]$, with different corresponding bonus function classes $\mathcal{W}\_{h} = \mathcal{W}\_{h, \mathcal{S} \times \mathcal{A}}$.*
>
> We hope this clarifies our intentions.
>
> ---
> **Q4.** How is an agent chosen to be activated in Line 5 of both Algorithm 1 and 2? What if there are more than one agents are to be activated instead of just having one agent at each time t?
>
> **A4.** Regarding the questions of the agents’ activation, even though we assume an order of activation for the agents in our work, it is only for the sake of convenience for our theoretical analyses. Realistically, agents individually decide when to activate and when to send its history data to the server and request policy updates, and only the communication order is visible to the server. Under realistic settings, the probability that two agents communicate at the exact same time is extremely low, and even under such events the server can simply process one request before moving on to the next, thus not affecting our overall algorithm.
>
> We modified the paragraph on Lines 204-206 in our main paper to the following for clarity:
>
> ***Part I: Local Exploration.** At step $t$ a single agent $m = m_t$ is active (Line 5). They receives a decision set, finds the greedy action according to its decision function $f\_{m, t}$, receives a reward, and updates its local dataset $Z\_{m, t} ^{\text{loc}}$ (Lines 5-7). Note that the specific order of agent activation does not affect our algorithm, as long as the communication order remains the same. This nicely reflects realistic fully asynchronous multi-agent settings, where each agent individually and independently interacts with the environment until communication is triggered by some criterion.*
>
> ---
> [1] Agarwal et al. VO$Q$L: Towards Optimal Regret in Model-free RL with Nonlinear Function Approximation. In Proceedings of Thirty Sixth Conference on Learning Theory, Jul 2023.
>
> [2] Zhao et al. A nearly optimal and low-switching algorithm for reinforcement learning with general function approximation, 2023.
>
> [3] Kong et al. Online sub-sampling for reinforcement learning with general function approximation, 2023.
>
> [4] Wang et al. Reinforcement learning with general value function approximation: Provably efficient approach via bounded eluder dimension. In Advances in Neural Information Processing Systems, 2020b.

---

> > ### Comment · Reviewer_56ui · 2024-08-09
> > **Response to rebuttal**
> >
> > I appreciate the authors' efforts on the response. I have one follow-up question with regard to A4. I am wondering the validity of the following scenario:
> >
> > In multi-agent setting, in particular when the number of agents is sufficiently large, the probability of two agents communicating with the server having an overlap time seems to be up to the switch condition. If the switch condition is easy to be satisfied, then the above probability is not negligible. In the case that an overlap does happen, an active sampling agent A who just satisfied the switch condition may have to wait for the server to finish processing the request (line 11 and 12) of another active agent B. The processing time may also take significant time depending on the new data size.
> >
> > Is above scenario possible to occur? Thanks.

---

> > > ### Author Response · Authors · 2024-08-10
> > > **Response to Follow-up Question**
> > >
> > > Thank you for the follow-up question!
> > >
> > > We first point out that in theory, we assume data upload, calculation of decision function $\hat{f}$ and bonus function $b$, as well as function download are all instantaneous (Lines 12-20 in Algorithm 2), so the proposed scenario will never occur and hence does not affect our theoretical analysis.
> > >
> > > That being said, should we implement our algorithm in practice, we believe this scenario is also very unlikely to happen for the following reasons:
> > > 1. As per our communication criterion in equation (8), each agent only communicates with the server once their local sum of uncertainty accumulates to $\alpha$, which means **the frequency of communication decreases exponentially** relative to round $K$. Therefore, when $K$ is large, the communication frequency becomes extremely low, which means the probability of overlap is also very low;
> > > 2. In order to avoid communication overhead when $K$ is small, one may consider initializing all local agents with some initial observations before deploying our asynchronous algorithm. This ensures that local agents will not start communicating until their observed data exceed a certain threshold.
> > >
> > > Finally, even without the assumption that communication is instantaneous, it is not difficult to modify our theoretical analysis to accommodate for overlapping communication and **delayed feedback**. If we simply ask the agent to **adhere to the old policy** until the updates are received, under proper assumptions the regret term will be incremented by $O(\log K)$ in the worst case. A large body of literature addresses optimization and bandits with delayed feedback, offering analysis techniques we could potentially leverage. However, extending our current methods in this direction is beyond the scope of our current work, therefore we do not consider it in our paper.

---

> > > > ### Comment · Reviewer_56ui · 2024-08-10
> > > > **Thank you for the response**
> > > >
> > > > The authors' detailed response has addressed my concern. The reviewer has revised the score.

---

### Decision · Program_Chairs · 2024-09-25

**Decision:**

Reject

**Comment:**

This paper studied asynchronous multi-agent reinforcement learning with general function approximation. Both regret and communication complexity have been analyzed, with solid theoretical derivations. However, there were concerns regarding the technical novelty of the results, and presentation/writing of the results. There were also some concerns regarding the tightness of the bounds. After reading the paper more carefully, I found the majority of the techniques and quantities follow closely from those in single-agent RL with general function approximation, which diminishes the novelty of the technical contributions. Moreover, the lack of experiments to validate the theories made the impact of the results relatively limited, beyond purely theoretical arguments. In general, it is a borderline paper, and the paper could benefit from the review feedback this round.